# ATraDiff: Accelerating Online Reinforcement Learning with Imaginary Trajectories

## Abstract

Training autonomous agents with sparse rewards is a long-standing problem in online reinforcement learning (RL), due to low-data efficiency. Prior work overcomes this challenge by extracting useful knowledge from offline data, often accomplished through the learning of action distribution from offline data and utilizing the learned distribution to facilitate online RL. However, since the offline data are given and fixed, the extracted knowledge is inherently limited, making it difficult to generalize to new tasks. We propose a novel approach that leverages offline data to learn a generative diffusion model, coined as *Adaptive Trajectory Diffuser (ATraDiff)*. This model generates synthetic trajectories, serving as a form of data augmentation and consequently enhancing the performance of online RL methods. The key strength of our diffuser lies in its adaptability, allowing it to effectively handle varying trajectory lengths and mitigate distribution shifts between online and offline data. Because of its simplicity, ATraDiff seamlessly integrates with a wide spectrum of RL methods. Empirical evaluation shows that ATraDiff consistently achieves state-of-the-art performance across a variety of environments, with particularly pronounced improvements in complicated settings.

## 1 Introduction

Deep reinforcement learning (RL) has shown great promise in various applications, such as autonomous driving (Wang et al., 2019), chip design (Mirhoseini et al., 2021), and energy optimization (Specht & Madlener, 2023). Despite its impressive performance, RL often requires extensive online interactions with the environment, which can be prohibitively costly in practice. Such downside of RL is aggravated by the fact that in real-world scenarios, environments are often characterized by sparse rewards, which further necessitates an exceptionally large number of samples for effective exploration; for example, when manipulating a robotic arm to move an item, oftentimes the only reward feedback given is at the success moment of the task, which may take hundreds of steps to obtain. Consequently, a persistent challenge in RL is addressing the high sample costs, particularly in contexts with sparse rewards.

One prevalent solution to this challenge is leveraging offline data, by directly learning policies from offline data (Kostrikov et al., 2022; Ball et al., 2023a) or extracting experiences that can enhance online training (Pertsch et al., 2020b), especially in the context of exploration. However, such solutions can only extract limited knowledge as the offline data are given and fixed, and thus are difficult to generalize to new tasks. In contrast to prior work, this paper takes a different perspective inspired by the recent advances in generative modeling – *can we harness modern generative models, such as diffusion models, trained on offline data and synthesize useful data that facilitate online RL?*

Indeed, diffusion models have emerged as powerful deep generative models, demonstrating impressive capabilities in data synthesis across vision and language applications (Ho et al., 2020; Gong et al., 2023; Li et al., 2022). Nevertheless, their investigation in RL has been relatively limited. Studies in offline RL mainly tap into the generative capabilities of diffusion models to enhance long-term planning (Janner et al., 2022; Ajay et al., 2023) and amplify policy expressiveness (Wang et al., 2023; Chen et al., 2023). Recently, Lu et al. (2023) have started to utilize diffusion models specifically for RL data augmentation, by upsampling the replay buffer of online RL with a diffusion model trained on offline data. However, such an approach primarily focuses on generating *transitions* rather than *complete trajectories*. This under-utilizes the generative potential of diffusion

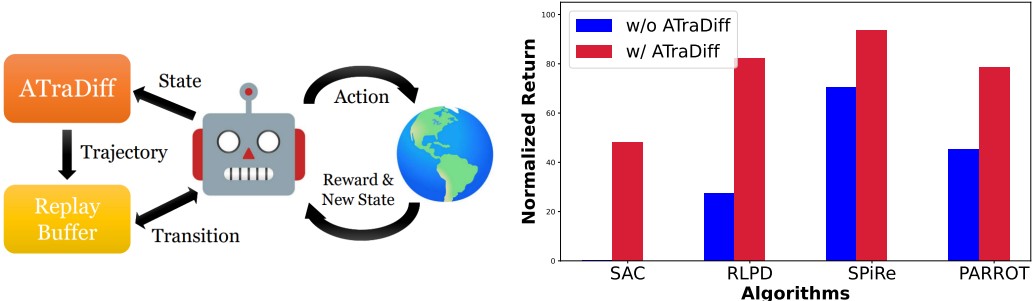

Figure 1: Illustration and performance showcase of our proposed ATraDiff. ATraDiff can seamlessly integrate with a wide range of RL methods and consistently improve their performance, by augmenting the replay buffer with synthesized trajectories. **Left**: Overview of online RL with ATraDiff. **Right**: Performance comparison of RL methods with and without ATraDiff in D4RL Kitchen.

models and limits the benefits of augmented data for RL. In contrast, full trajectories offer a more comprehensive source of information, enabling RL agents to better learn from past experiences.

To overcome these issues, we propose *Adaptive Trajectory Diffuser (ATraDiff)*, a novel method designed to synthesize full trajectories for online RL. As depicted in Figure 1, our approach trains a diffusion model using offline data, which then synthesizes complete trajectories conditioned on the current state. By employing this diffusion model to produce additional trajectories, we aim to significantly accelerate the online RL process. Notably, because of its simplicity in augmenting the replay buffer by adding useful data, ATraDiff *seamlessly integrates with a wide range of RL methods* and *consistently* elevates their performance.

The key property of our diffuser lies in its *adaptability* to effectively handle varying trajectory lengths and address the distribution shifts between online and offline data. Unlike generating transitions, managing the uncertainty in task length presents a significant new challenge in trajectory generation. While longer trajectories can potentially lead to improved performance, excessive or redundant segments may be detrimental. Ideally, we aim for a generation with precise trajectory length. To this end, we introduce a simple *coarse-to-precise* strategy: initially, we train multiple diffusion models with varying generation lengths. Prior to actual generation, we assess the required length and subsequently prune any redundant segments. In dealing with the distribution shift between offline data and online evaluation tasks, we design our diffuser to be adaptable throughout the RL process. This adaptability includes the capability to select more informative samples through the use of an *importance indicator*, while also mitigating catastrophic forgetting during adaptation.

**Our contributions** are three-fold. **(i)** We propose ATraDiff, a novel diffusion-based approach that leverages offline data to generate full synthetic trajectories and enhance the performance of online RL methods. ATraDiff is general and can be seamlessly applied to accelerate *any* online RL algorithms with a replay buffer. **(ii)** We introduce a simple coarse-to-precise strategy that ensures generated trajectories precisely align with the length required for evaluation tasks. **(iii)** We devise an online adaptation mechanism that effectively addresses challenges stemming from data distribution shifts. Empirical evaluation shows that ATraDiff consistently achieves state-of-the-art performance across a variety of environments, with particularly large improvements in complicated settings.

## 2 RELATED WORK

**Offline pre-training for online RL.** Leveraging prior experience to expedite online learning for subsequent tasks has been a persistent challenge (Nair et al., 2021). Past research has proposed numerous solutions to tackle this issue. Some studies suggest treating offline data similarly with data collected online. For instance, representative approaches employ offline data to initialize a replay buffer (Večerík et al., 2017; Hester et al., 2018). Meanwhile, others advocate for a balanced sampling strategy, drawing from both offline and online sources (Nair et al., 2018; Kalashnikov et al., 2018; Hansen et al., 2022; Zhang et al., 2023; Ball et al., 2023a).

One prevalent approach is to establish a behavior prior, which captures the action distribution in prior experiences to mitigate overestimation for actions outside the training data distribution (Singh et al., 2020; Siegel et al., 2020). An alternative strategy involves extracting skills from offline data and adapting them to new tasks (Gupta et al., 2019b; Merel et al., 2019; Kipf et al., 2019; Whitney et al., 2020; Pertsch et al., 2020b). These studies typically represent the acquired skills within an embedding space and then train a policy to select the most appropriate skills based on the current state. In contrast, ATraDiff synthesizes the complete trajectory based on the current state and augments the replay buffer with these additional data. Our approach offers broader applicability across a diverse set of RL methodologies.

**Diffusion models in RL.** Previously, the focus of employing diffusion models in RL has primarily centered on enhancing long-term planning and amplifying policy expressiveness. For instance, Diffuser (Janner et al., 2022) constructs a full trajectory of transitions, through conditioned sampling guided by higher rewards and goal-oriented navigation. This leverages the diffusion model's capability in generating extensive trajectories, addressing challenges such as long horizons and sparse rewards in RL planning. Similarly, several other studies (Ajay et al., 2023; Du et al., 2023; He et al., 2023) have adopted this paradigm, particularly in the context of visual data. Another notable work (Pearce et al., 2023) suggests a diffusion-based method for generating full trajectories by imitating human behavior. Different from prior work, our approach is oriented towards synthesizing trajectories in evaluation scenarios that may differ from the training scenarios.

**Data augmentation in RL.** Data augmentation is a common technique that has demonstrated effectiveness in RL. Previous methods (Yarats et al., 2021; Laskin et al., 2020; Sinha et al., 2021) typically focus on perturbing original data on observations for visual-based RL, such as adding noise and applying random translation. This enables agents to learn from multiple views of the same observation and increase their robustness. Different from prior work on diffusion models in RL, recent efforts have focused on upsampling the replay buffer with the diffusion model. A closely related work by Lu et al. (2023) generates transitions to augment the replay buffer via a diffusion model. However, our ATraDiff *operates at the trajectory level, employing a visual-based diffusion model, and has the capability to synthesize training data through state and task-conditioned generation.*

## 3 BACKGROUND

**MDP.** In this paper, we consider sequential decision-making tasks that can be modeled as a Markov Decision Process (MDP) defined as $\mathcal{M} = \langle S, A, T, R, \gamma \rangle$, where $S$ is the set of states, $A$ is the set of actions, and $\gamma \in [0, 1)$ is the discount factor. $T(s'|s, a)$ and $R(s, a)$ represent the dynamics and reward functions, respectively. At each stage $t$, the agent takes an action $a \in A$, which leads to a next state $s'$ according to the transition function $T(s'|s, a)$ and an immediate reward $R(s, a)$. A trajectory of such a task is defined as a sequence composed of states and actions given by $(s_1, a_1, s_2, a_2, \ldots, s_t, a_t)$, where $s_t$ and $a_t$ denote the state and action at time-step $t$, respectively.

**Diffusion models.** Diffusion probabilistic models pose the data-generating process as an iterative denoising procedure $p_\theta(\tau^{i-1}|\tau^i)$. This denoising is the reverse of a forward diffusion process $q(\tau^i|\tau^{i-1})$ that slowly corrupts the structure in data by adding noise. The data distribution induced by the model is given by:

$$p_\theta(\tau^0) = \int p(\tau^N) \prod_{i=1}^{N} p_\theta(\tau^{i-1}|\tau^i) d\tau^{1:N},$$

where $p(\tau^N)$ is a standard Gaussian prior and $p(\tau^0)$ denotes noiseless data. Parameters $\theta$ are optimized by minimizing a variational bound on the negative log-likelihood of the reverse process: $\theta^* = \arg\min_\theta -\mathbb{E}_{\tau^0}[\log p_\theta(\tau^0)]$. The reverse process is often parameterized as Gaussian with fixed timestep-dependent covariances:

$$p_\theta(\tau^{i-1}|\tau^i) = \mathcal{N}(\tau^{i-1}|\mu_\theta(\tau^i, i), \Sigma^i).$$

**Replay buffers.** Many algorithms in RL employ replay buffers, denoted as $D$, to retain trajectories derived from executing a sample policy within an environment parameterized by MDP. Throughout the training process, these replay buffers are accessed to extract samples (transitions or trajectories) for the update of the learned execution policy.

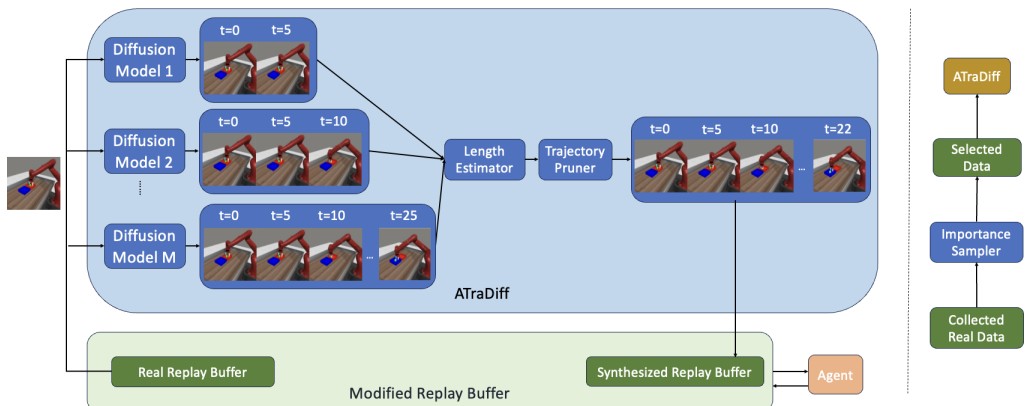

Figure 2: An illustrative overview of ATraDiff framework. **Left**: A generator containing multiple diffusers, a length estimator, and a trajectory pruner. **Right**: Workflow of the online adaptation.

## 4 METHOD

We now present our approach to accelerating online reinforcement learning by training our generative model ATraDiff on the offline data to synthesize trajectories. We begin by introducing how we design and train ATraDiff (Sec. 4.1) and then how we apply ATraDiff to accelerate online reinforcement learning (Sec. 4.2). Finally, we will introduce how this generator could be dynamically adapted during the online training process (Sec. 4.3). Figure 2 illustrates the overall framework of our approach.

### 4.1 ADAPTIVE TRAJECTORY DIFFUSER

Our primary objective with the generator is to train a diffusion model $p(x)$ that captures the trajectory data distribution present within the offline data. By doing so, we can synthesize new trajectories $(s_1, a_1, s_2, a_2, \ldots, s_t, a_t)$ with the learned diffusion model. To harness the diffusion model more effectively, we advocate for the synthesis of trajectory *images* rather than direct generation of states and actions. Our strategy entails initially generating images $m_1, m_2, m_3, \ldots, m_k$ spanning continuous $k$ frames, where $k$ is a fixed generation length preset for a given generator. Subsequently, we can extract the state $s_i$ from the resultant images using an environment-specific *recognizer* that converts images into states using object detection, and retrieve actions from adjacent states using physical simulation. Conversely, to provide ground-truth images for our generator, we use another *renderer* to convert states into images. We implement the *recognizer* by detecting the 3D position of key points in the figure, including points of the object boundary and the junctura of the robot arm. And the *renderer* is implemented by simple ray tracing (Shirley, 2000).

**Image diffusion models.** For the architecture design of our image diffuser, we use Stable Diffusion (Rombach et al., 2022) as the pretrained model and fine-tune it on the specific dataset. To generate images of $k$ continuous frames, we concatenate the $k$ single images $m_1, m_2, \ldots, m_k$ into one larger 2D image $M$ by aligning them in order and use the diffusion model to generate the concatenated image $M$. Furthermore, we use the information of the first frame as the generation condition, including the current state, task, etc., so that the generated trajectory will be closely related to our learning process, where our online learning could benefit more from those synthesized data.

**Flexible task-length generation.** Due to the property of diffusion models, our diffuser described above could only get trajectories with fixed length $k$; however, in order to generate a complete trajectory where the horizon is indefinite, we would require the diffuser to output trajectories with flexible lengths, which becomes the crucial problem. To solve this, we introduce a simple *coarse-to-precise* strategy. We initially train multiple diffusion models regarding different lengths $k = 5, 10, 15, \ldots$. Before generation, we first estimate the possible length we need for the current state

---

**Algorithm 1** Modified Replay Buffer for RL

---

**Require:** $D = (D_s, D_o, \rho, L)$, ATraDiff.

1: **function** STORE($D$, $z = (s, a', s', r)$)
2:     ReplayBufferStore($D_o$, $z$)
3:     **if** with probability $\frac{\rho}{(1-\rho)L}$ **then**
4:         $(s_1, a_1, \ldots, s_t, a_t) \leftarrow$ ATraDiff($s'$)
5:         **for** $\forall i$ **do**
6:             $z_i = (s_i, a_i, s'_i, R(s_i, a_i))$
7:             ReplayBufferStore($D_s$, $z_i$)
8:         **end for**
9:     **end if**
10: **end function**

11:
12: **function** SAMPLE($D$)
13:     **if** with probability $\rho$ **then**
14:         $z \leftarrow$ ReplayBufferSample($D_s$)
15:     **else**
16:         $z \leftarrow$ ReplayBufferSample($D_o$)
17:     **end if**
18:     **return** $z$
19: **end function**
20:
21:

---

with a pre-trained network and use the diffuser with the closest generation length, minimizing the length of redundant transitions to cut. To cut this part after generation, we use a non-parametric algorithm to find the best ending position.

Specifically, we first pad the training data: given a full trajectory $m_1, m_2, \ldots, m_t$, we will pad $k$ more frames identical to the last frame $m_t$ after the end of the trajectory, so it becomes $m_1, m_2, \ldots, m_t, m_t, m_t, \ldots, m_t$. Hence for the sub-trajectories starting after time $t - k$, the last part of it should always be the same. For example, for data starting with $t - 3$, the data should be $m_{t-3}, m_{t-2}, m_{t-1}, m_t, m_t, m_t, \ldots$. Therefore, in our generated result, the redundant part tends to become some similar states. For a generated trajectory, we first calculate the similarity between each of two adjacent states, $\text{sim}(s_1, s_2), \text{sim}(s_2, s_3), \ldots, \text{sim}(s_{k-1}, s_k)$. Then we compute the prefix average $\text{pre}_i$ and suffix average $\text{suf}_i$ of this similarity sequence, where $\text{pre}_i = \frac{\sum_{j=1}^{i} \text{sim}(s_j, s_{j+1})}{i}, \text{suf}_i = \frac{\sum_{j=i}^{k} \text{sim}(s_{j-1}, s_j)}{t-i+1}$. Then we get the difference between the prefix average and suffix average $\text{pre}_i - \text{suf}_i$ of each position and find the one with the largest difference to be the ending position. This algorithm works based on the fact that the average similarity before the ending point should be significantly larger than the one after the ending point.

## 4.2 DIFFUSER DEPLOYMENT

With the diffuser of ATraDiff explained, we now delve into how the diffuser can be seamlessly integrated with any online RL method with a replay buffer. Intuitively, ATraDiff augments the replay buffer with the data synthesized by its diffuser, and leaves the RL algorithm itself untouched; thus, our approach is orthogonal to any online RL method equipped with a replay buffer.

More specifically, consider any RL algorithm with a replay buffer denoted as $D_o$. Typically, RL methods engage with the replay buffer through two primary actions: *store* - which archives a new transition into the replay buffer, and *sample* - which extracts a transition randomly from the replay buffer. With this in mind, we substitute the original buffer $D_o$ with a new replay buffer $D = D_o \cup D_s$, where $D_s$ is the augmenting buffer synthesized by ATraDiff. This modified replay buffer $D$ is characterized by two hyperparameters: $\rho \in [0, 1]$, denoting the probability of sampling from synthesized data $D_s$ in RL, and $L \in \mathbb{N}$, indicating the expected length of synthesized trajectories.

Whenever we *store* a transition $(s, a, s', r)$ into $D$ during the RL algorithm, the following three steps are performed:

- We first *store* it into $D_o$;

- With probability $\frac{\rho}{(1-\rho)L}$, a full trajectory $(s_1, a_1, s_2, a_2, \ldots, s_t, a_t)$ is synthesized with the diffuser, with state $s'$ as the initial state (i.e., $s' = s_1$). Note the probability $\frac{\rho}{(1-\rho)L}$ is designed to keep the ratio between the total size of $D_s$ and $D_o$ to be $\frac{\rho}{1-\rho}$;

- All synthesized transitions $(s_i, a_i, s_{i+1}, R(s_i, a_i))$ will be *stored* to the replay buffer $D_s$, where the reward $R(s_i, a_i)$ is calculated from the environment.

When we *sample* from $D$, we *sample* from $D_s$ with probability $\rho$ and from $D_o$ with probability $1 - \rho$. See Alg. 1 for pseudo-code. Note the sample process can be arbitrary, i.e., our method is also orthogonal to other sampling techniques such as prioritized buffer (Schaul et al., 2016).

### 4.3 ONLINE ADAPTATION

Although the fixed ATraDiff can improve the performance of RL methods in some simple environments, we may still face the problem of distribution shift between the evaluation task and the offline data in complicated environments. To overcome this issue and further improve the generation quality, we propose an online adaptation technique by continually training the diffusion model on new experiences.

Concretely, ATraDiff is periodically updated on the real transitions stored in $D_o$ and then used to generate new trajectories. Meanwhile, we will keep a copy of the original version of the diffuser to mitigate potential catastrophic forgetting.

Furthermore, we will use more valuable samples to adapt our ATraDiff during online training. Specifically, we design some *indicators* to measure the importance of each sample for our online learning, and a pick-up strategy to choose samples from $D_o$. By default, we are introducing two indicators, the TD-error indicator, and the Reward indicator. For the TD-error indicator, the importance of a transition $(s, a, s', r)$ is defined to be $|r + \gamma \max_{a'} Q(s', a') - Q(s, a)|$. The TD-error indicator performs better in most cases, while it could only be used in some value-based RL methods. The Reward indicator would be more general to all RL methods, as the importance of a transition is defined to be the total reward collected in the full trajectory. The primary pick-up strategy is to maintain a subset of samples with higher importance, but part of the samples are always used to update the diffuser. Hence, we will randomly drop some samples from the maintained subset regardless of their importance. We further conduct experiments to analyze the effectiveness of our indicator and pick-up strategy and the effect of different design choices in the ablation study.

## 5 EXPERIMENTS

In this section, we conduct comprehensive experiments to evaluate the effectiveness of our data generator ATraDiff. First, we validate that our approach is able to improve the performance of both basic and state-of-the-art online RL methods by combining them with our ATraDiff (Sec. 5.1). Next, we show that our method can further improve the performance of a variety of state-of-the-art offline-to-online RL algorithms in complicated environments using online adaptation. Finally, we conduct some ablation studies to validate the effectiveness of different components in our approach. For evaluation, all results in this section are presented with the median performance over 5 random seeds as well as the 25%-75% percentiles.

### 5.1 ATRADIFF IMPROVES ONLINE RL

In this section, we show that the performance of state-of-the-art online RL methods can be improved by our ATraDiff learned with the offline data. We consider 3 environments from D4RL Locomotion (Fu et al., 2020), including 12 different offline data with varying levels of expertise. For comparison, we choose SAC (Haarnoja et al., 2018) as the basic online RL algorithm and REDQ (Chen et al., 2021) as state-of-the-art sample-efficient algorithm. We run two baselines and the versions combined with our ATraDiff for 250K steps.

The overall result is summarized in Figure 3. We see that the performance of the two baselines is both significantly improved by our ATraDiff, especially on the halfcheetah and walker2d environments. This validates the strength of ATraDiff. If we run these online RL methods for enough time, they can also achieve comparable results, while ATraDiff improves the sample efficiency. Here, we only use the fixed diffuser instead of using online adaption, which indicates that our fixed diffuser could already be used to accelerate the online RL in some simple environments.

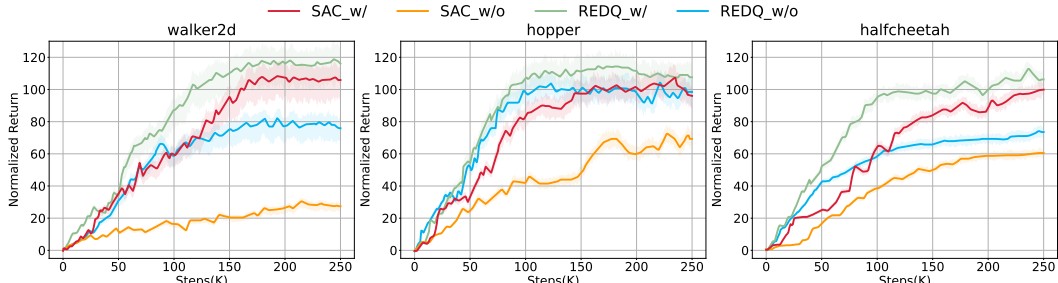

Figure 3: Learning curves on the D4RL Locomotion benchmark. ATraDiff (denoted as 'w/') consistently and significantly improves the performance of the two representative RL methods across all three environments, irrespective of whether basic or advanced algorithms are employed. These results validate the effectiveness and generalizability of our diffuser.

## 5.2 ATRADIFF IMPROVES OFFLINE-TO-ONLINE RL IN COMPLICATED ENVIRONMENTS

In this section, we show that ATraDiff with online adaptation could be used to improve the performance of state-of-the-art offline-to-online RL methods. We further consider the following environments:

- D4RL AntMaze (Fu et al., 2020). There are 6 sparse reward tasks that require the agent to learn to walk with controlling an 8-DoF Ant robot and navigate through a maze to reach the goal.

- D4RL Kitchen (Fu et al., 2020). A simulated kitchen environment first proposed by Gupta et al. (2019a), which involves controlling a 9-DoF robot that manipulates different objects in a kitchen environment (e.g., slide cabinet door, switch overhead light, open microwave). The downstream task is to complete a sequence of multiple subtasks in order with a sparse, binary reward for each successfully completed subtask. The offline dataset only contains part of the full sequence, meaning that the agent needs to learn to compose sub-trajectories.

- Meta-World Environment (Yu et al., 2019). By combining the modified tasks using a single camera viewpoint consistently over all the 15 subtasks generated by Seo et al. (2022), we create two challenging tasks. The first one is a multi-task setting, where the offline data contain the trajectories of 14 subtasks and evaluates on the remaining 1 subtask. The second one is a harder version of the D4RL Kitchen task, where the agent needs to complete a sequence of 8 subtasks with medium difficulty in the correct order, while the offline data only contain trajectories for single subtasks. For both tasks, the trajectories of any single subtask are consisted the provided data and collected data from the online training.

We consider a set of strong baselines from prior work on offline-to-online RL, including skilled-based method (SPiRL (Pertsch et al., 2020a)), behavior-prior-based method (PARROT (Singh et al., 2021)), balanced sampling method (RLPD (Ball et al., 2023b)). We run these baselines and the versions combined with our diffuser.

As shown in Figure 4, our diffuser could significantly improve the performance of some methods, and get at least comparable results for all these methods. For all these environments, the best one is always some method combined with our diffuser, which shows that our online adapted diffuser could further improve the performance of current offline-to-online methods. Moreover, we introduce two advanced tasks within the Meta-World environment to accentuate the challenges posed by distribution shifts. Our motivation is to understand how our method responds under conditions where distribution shift problems are more pronounced. As demonstrated in Figure 5, our approach exhibits superior performance, suggesting that our method holds significant promise in effectively addressing such shifts.

## 5.3 ABLATION STUDY AND ANALYSIS

In this section, we carry out ablation studies to justify the effect of different components of our ATraDiff.

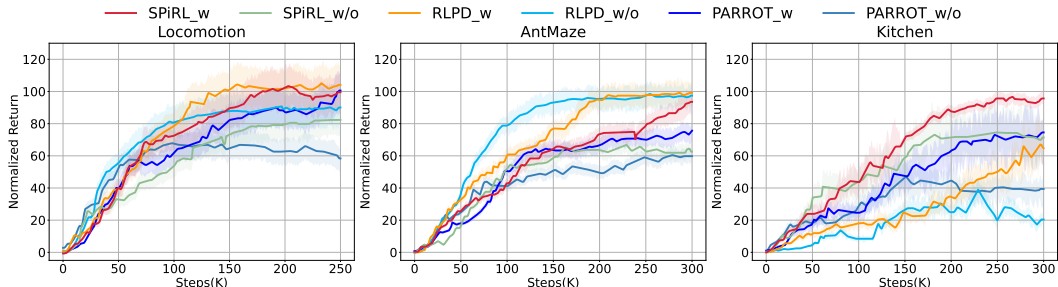

Figure 4: Learning curves on the D4RL benchmark. ATraDiff (denoted as 'w/') further boosts the performance of advanced and recent offline-to-online RL baselines across all three environments, leading to state-of-the-art results especially in complex settings, where the improvements are particularly noteworthy. This shows the importance of our online adapted diffuser.

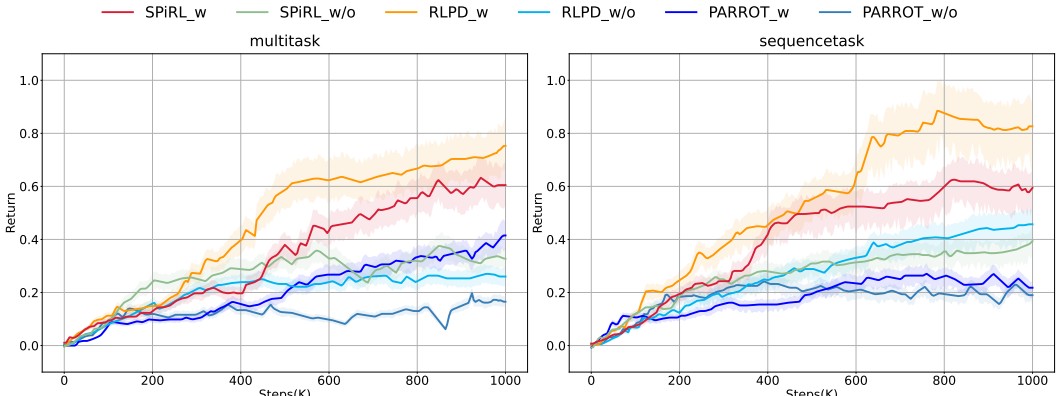

Figure 5: Learning curves on the Meta-World benchmark. While the two tasks within the Meta-World environment are designed purposefully to be very changeling with considerable distribution shifts, ATraDiff (denoted as 'w/') is still effective and significantly improves the performance of advanced and recent offline-to-online RL baselines. This further validates the strength of ATraDiff in tacking distribution shifts between offline data and online tasks.

**Image-level generation vs. state-level generation.** We now illustrate the importance of image-level diffuser by comparing it with the results conducted by a state-level diffuser. As shown in Figure 6, the image-level diffuser generally outperforms the state-level diffuser on the D4RL Locomotion environment (Fu et al., 2020). We usually need to synthesize longer and more flexible trajectories in such complicated environments, where the image-level diffuser performs better compared with simpler diffusion models.

**Is online adaptation beneficial for our diffuser?** We now examine the effect of online adaptation on performance. We revisit the experiments in Section 5.1 and replace the fixed diffuser with an online adapted one. As shown in Figure 7, we found that the online diffuser significantly outperforms the fixed diffuser in complicated tasks. In tasks of D4RL Locomotion and D4RL AntMaze (Fu et al., 2020), the online diffuser

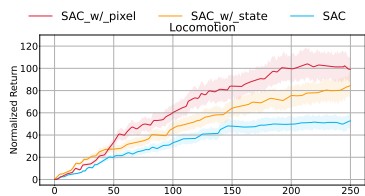

Figure 6: Ablation study on image-level generation and state-level generation on the D4RL Locomotion Environment. The image-level diffuser outperforms the state-level diffuser in complicated tasks, with noticeable performance gains.

achieves comparable results to the fixed diffuser. However, in task of D4RL Kitchen and two tasks in Meta-world (Yu et al., 2019), the online diffuser has fully demonstrated its superiority, which validates that the online adaptation can indeed mitigate the problem of data distribution shift and focus on the evaluation task.

**Effect of different importance indicators over online adaptation.** Here we show that different online adaptation strategies have noticeable impacts on performance. We conduct experiments to test

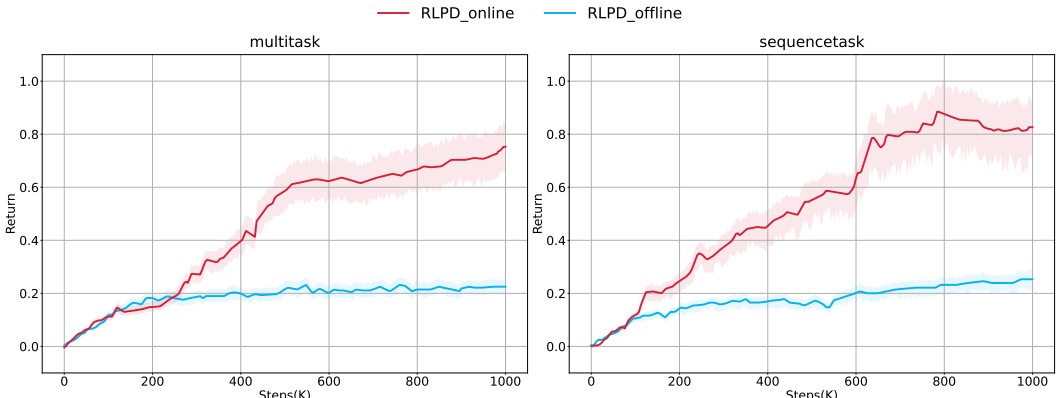

Figure 7: Ablation study on online adaptation of ATraDiff. In simple tasks, we observe that the on-line diffuser achieves results comparable to the fixed diffuser. However, online adaptation mitigates the problem of data distribution shift and becomes critical in complicated environments.

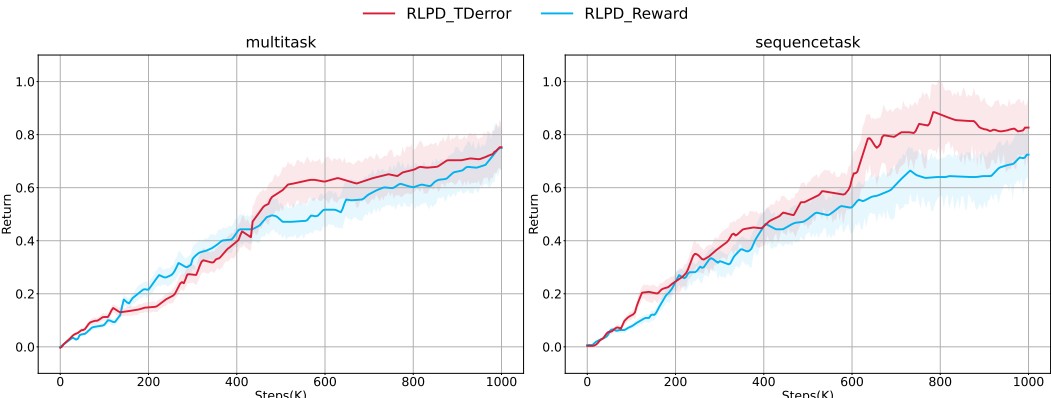

Figure 8: Ablation study on the importance indicator together with its associated pick-up strategy. Different types of indicators lead to varied behavior and performance. While the Reward indicator is in principle more general to RL methods, our indicator based on TD-error achieves better perfor-mance in the complicated Meta-World benchmark.

the effect of different design choices in the online adaptation, focusing on the indicator used to mea-sure the importance of collected samples together with its associated pick-up strategy determining the samples according to the importance. We include the Reward indicator and its corresponding pick-up strategy. The result shown in Figure 8 demonstrates that different choices of the online adaptation indeed affects the performance.

## 6 CONCLUSION

This paper introduces ATraDiff, a novel diffusion-based approach that synthesizes full trajectories. By training a set of diffusion models that generate different lengths of trajectories and selecting the most suitable model with non-parametric similarity comparison, we obtain a generator that pro-duces trajectories with varied lengths. By applying such a generator to transitions stored in the replay buffer and augmenting the buffer with generated data, ATraDiff can accelerate any online RL algorithms with a replay buffer using imaginary data. In multiple environments, we found that ATraDiff significantly improves existing online and offline-to-online RL algorithms. We believe that our work is useful to the online RL community on the long-standing data efficiency problem.

**Limitations and future work.** One limitation of ATraDiff is that the training of multiple diffusion models can be computationally intensive, and the problem grows with longer generated trajectories. Thus, an interesting future direction is to learn a single diffuser that directly generates trajectories of varied lengths instead of selecting from a collection of models.

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

# A    ALGORITHM DETAILS

## A.1    PICK-UP STRATEGIES FOR THE ONLINE ADAPTATION

In our training methodology, we have designed different pick-up strategies for handling different importance indicators, recognizing their unique properties and implications for the learning process.

For the TD-error indicator, a dynamic approach is employed due to the variable nature of its importance, which can change as the critic function updates during training. To manage this, a heap structure is used to maintain a buffer of 50,000 samples. This buffer is continuously updated to reflect the current importance of each sample, ensuring that even those with initially low importance are retained for potential future relevance. When it comes time to update the diffuser, a selection of 5,000 high-importance samples is drawn from this buffer, aligning the update process with the most pertinent data at that moment.

In contrast, the strategy for reward importance takes into account its static characteristic – the importance of these indicators does not change throughout the training. Here, a smaller buffer of 5,000 samples is employed, updated similarly based on importance. However, to counter the risk of high-importance samples perpetually dominating the buffer, a rotation system is implemented. After each update, some samples, particularly the older ones, are dropped. This system ensures that each sample contributes to the diffuser update only a limited number of times, thus maintaining a fresh and current dataset for training, reflective of the latest environmental interactions.

These tailored strategies highlight a nuanced understanding of how different indicators behave and affect the learning process, ensuring that both dynamic and static aspects of the training data are optimally utilized for updating the diffuser.

## A.2    STATE-LEVEL TRAJECTORY GENERATION

In the state-level generation, we directly generate trajectories consisting of states and actions, $\{s_t, a_t, s_{t+1}, a_{t+1}, \dots\}$. For the architecture of the diffusion model used in the state-level generation, we directly refer to the architecture used in Lu et al. (2023), while we omit the generation of the reward. Meanwhile, we extend the size of the network used from 24 to 128 to support the larger result of a trajectory.

## A.3    RENDERER AND RECOGNIZER

In the image-level generation, a renderer and a recognizer are employed for image rendering and key point detection tasks, respectively. The renderer, based on previous work (Lu et al., 2022; Hansen et al., 2022; Seo et al., 2022), captures observations in an environment where an AI agent operates. The images rendered, set at a resolution of $84 \times 84$, serve as input of the generation model.

The recognizer is a neural network consisting of two components. The first component utilizes a CNN to extract features from the images. In the second component, separate 5-layer MLP networks are used to determine the 3D positions of specific key points, which are selected based on the requirements of different tasks. This two-component design of the recognizer allows for efficient processing of the images, first by identifying relevant features through CNN and then pinpointing key points' positions using MLP networks, so that we can calculate the proprioceptive state.

# B    DETAILS OF EXPERIMENT SETUP

## B.1    D4RL

We basically consider 3 different environments from D4RL (Fu et al., 2020). We use the original offline dataset from D4RL (Fu et al., 2020).

**AntMaze**. This domain steps up the complexity by replacing the 2D ball in Maze2D with an 8-DoF "Ant" quadruped robot, adding a layer of morphological complexity. It is a navigation domain that closely resembles real-world robotic navigation tasks. We followed the design of a sparse 0-

| Task Name | Subtasks |
|---|---|
| Multitask | evaluation task: Push |
| Sequencetask | Sweep, Sweep Into, Coffee Push, Box Close, Push Wall, Peg Insert Side, Basketball, Soccer, |

Table 1: **Meta-World task setting.** We select push task as the evaluation task for the multitask environment. We arrange all the 8 *medium* to form the Sequencetask environment.

1 reward system in this environment, activated only upon reaching the goal, to test the stitching challenge under more complex conditions.

**Locomotion**. Comprising tasks like Hopper, HalfCheetah, and Walker2d, the Locomotion domain is a staple in offline deep RL benchmarks. We used the same datasets in D4RL (Fu et al., 2020) for consistency with previous studies, and also experimented with a variety of datasets to observe the effects of different data qualities.

**Kitchen**. This environment involves controlling a 9-DoF Franka robot in a kitchen setting, interacting with everyday household items like a microwave, kettle, cabinets, an overhead light, and an oven. Each task aims to achieve a specific goal configuration, like opening the microwave and sliding cabinet door, placing the kettle on the burner, and turning on the overhead light. The Kitchen domain serves as a benchmark for multitask behavior in a realistic, non-navigation setting. Here, the stitching challenge is amplified due to the complexity of the trajectories through the state space, compelling algorithms to generalize to unseen states rather than rely solely on training trajectories.

### B.2 META-WORLD

The Meta-World benchmark (Yu et al., 2019) is a comprehensive suite designed for evaluating and advancing reinforcement learning and multi-task learning algorithms. It features 50 distinct robotic manipulation tasks, offering a diverse and challenging environment for testing the ability of algorithms to generalize and quickly acquire new skills. By providing a broad range of tasks, Meta-World seeks to address the limitations of existing benchmarks that focus on narrow task distributions. This benchmark is instrumental in fostering research that moves towards meaningful generalization, enabling algorithms to effectively learn multiple tasks and adapt to entirely new behaviors, which is crucial for the practical application of reinforcement learning in dynamic real-world scenarios. Following previous work (Hansen et al., 2022), we selected a total of 15 tasks from Meta-World based on their difficulty according to one previous work (Seo et al., 2022), which categorizes tasks into *easy*, *medium*, *hard*, and *very hard* categories. Same as Hansen et al. (2022), we discard *easy* tasks and select all tasks from the remaining 3 categories. Our approach differs from previous studies (Hansen et al., 2022; Seo et al., 2022) in that we solely utilize proprioceptive state information as input, rather than RGB frames or combined state information. This choice allows for easier application to general reinforcement learning methods. Following the previous settings, we use a sparse-reward signal that only provides a reward of 1 when the current task is solved and 0 otherwise. For the success criteria, we follow the original setting in Meta-World (Yu et al., 2019). Table 1 offers a detailed setting of our multitask and sequence task.

**Training Dataset**. The dataset is acquired by training an RL agent with SAC on each single task. For the multi-task setting, the entire dataset is combined by all 14 single tasks. For the sequence task, the entire dataset is combined by all 8 single tasks, without any trajectory change.

## C ADDITIONAL EXPERIMENTAL RESULTS

This section includes additional experimental results, which are ablation studies and detailed results that help to better understand the properties of different methods and components.

### C.1 OFFLINE EXPERIMENTS

In this section, we present experiments designed to evaluate the performance of our ATraDiff model in an offline setting.

**Comparison with transition-level generation SynthER.** Our objective is to verify whether ATraDiff can match or surpass the results achieved in previous research, specifically referencing the work SynthER (Lu et al., 2023). To ensure a fair and accurate comparison, we meticulously replicated the experimental settings used in Lu et al. (2023).

Our testing ground is the D4RL Locomotion benchmark, as detailed in (Fu et al., 2020). We also extend the original dataset to $5M$ samples, following the settings in Lu et al. (2023).

The findings, as outlined in Table 2, are quite revealing. They indicate that our ATraDiff method generally outperforms SynthER across various datasets. This performance improvement is particularly pronounced in high-quality datasets, underscoring the effectiveness of our approach in generating helpful trajectories and the benefits of trajectory-level generation over transition-level generation.

**Learned reward predictor.** Furthermore, we introduce a reward predictor to avoiding accessing the ground-truth reward function. Following previous work (Konyushkova et al., 2020), we use a supervised learning from demonstrations, by minimizing the loss:

$$L_{\sup}(D_0) = \mathbb{E}_{s_t \sim D_0}[r_t - R(s_t, a_t, s_t')]^2,$$

where $D_0$ represent the demonstration dataset with reward label, $(s_t, a_t, s_t')$ is the state with reward label $r_t$, and $R(s_t, a_t, s_t')$ is the learned reward function.

Results shown in Table 3 illustrate that with the reward predictor, our method could still improve the performance of offline RL methods, but slightly worse than the result with ground-truth reward.

| Task Name | TD3+BC | TD3+BC+SynthER | TD3+BC+ours | IQL | IQL+SynthER | IQL+ours |
|---|---|---|---|---|---|---|
| halfcheetah-random | 11.3 | 12.2 | 12.5 | 15.2 | 17.2 | 17.1 |
| halfcheetah-medium | 48.1 | 49.9 | 52.3 | 48.3 | 49.6 | 53.1 |
| halfcheetah-replay | 44.8 | 45.9 | 46.5 | 43.5 | 46.7 | 49.2 |
| halfcheetah-expert | 90.8 | 87.2 | 93.6 | 94.6 | 93.3 | 95.2 |
| hopper-random | 8.6 | 14.6 | 15.2 | 7.2 | 7.7 | 8.1 |
| hopper-medium | 60.4 | 63.4 | 65.7 | 62.8 | 72.0 | 72.4 |
| hopper-replay | 64.4 | 53.4 | 64.7 | 84.6 | 103.2 | 103.6 |
| hopper-expert | 101.1 | 105.4 | 111.2 | 106.2 | 90.8 | 113.6 |
| walker-random | 0.6 | 2.3 | 2.1 | 4.1 | 4.2 | 4.3 |
| walker-medium | 82.7 | 84.8 | 87.5 | 84.0 | 84.7 | 89.1 |
| walker-replay | 85.6 | 90.5 | 86.3 | 82.6 | 83.3 | 85.4 |
| walker-expert | 110.0 | 110.2 | 111.2 | 111.7 | 111.4 | 111.7 |

Table 2: Results of the offline experiments on the D4RL Locomotion benchmark (Fu et al., 2020). We show that ATraDiff outperforms SynthER on almost all the tasks and dataset.

| Task Name | TD3+BC | TD3+BC+ours | TD3+BC+ours_w/ _learned_reward | IQL | IQL+ours | IQL+ours_w/ _learned_reward |
|---|---|---|---|---|---|---|
| halfcheetah-random | 11.3 | 12.5 | 11.7 | 15.2 | 17.1 | 16.8 |
| halfcheetah-medium | 48.1 | 52.3 | 49.3 | 48.3 | 53.1 | 50.2 |
| halfcheetah-replay | 44.8 | 46.5 | 45.3 | 43.5 | 49.2 | 45.6 |
| halfcheetah-expert | 90.8 | 93.6 | 91.9 | 94.6 | 95.2 | 94.5 |
| hopper-random | 8.6 | 15.2 | 14.8 | 7.2 | 8.1 | 8.3 |
| hopper-medium | 60.4 | 65.7 | 64.5 | 62.8 | 72.4 | 72.2 |
| hopper-replay | 64.4 | 64.7 | 65.2 | 84.6 | 103.6 | 97.0 |
| hopper-expert | 101.1 | 111.2 | 108.7 | 106.2 | 113.6 | 108.3 |
| walker-random | 0.6 | 2.1 | 3.4 | 4.1 | 4.4 | 4.0 |
| walker-medium | 82.7 | 87.5 | 85.0 | 84.0 | 89.1 | 85.3 |
| walker-replay | 85.6 | 86.3 | 87.5 | 82.6 | 85.4 | 84.9 |
| walker-expert | 110.0 | 111.2 | 111.1 | 111.7 | 111.7 | 111.6 |

Table 3: Ablation study on the reward predictor. We show that with the learned reward predictor, our ATraDiff can still improve the performance of the offline RL methods.

## C.2  ABLATION STUDIES

**Effect of different task prompts.** Here we show that different task prompts have noticeable impacts on performance. We conduct experiments on the Meta-World environments (Yu et al., 2019) to test the effect of different design choices for the task prompts, focusing on how to represent the task. We

include three different strategies, language task prompt, one-hot task prompt, and no prompt, where the one-hot prompt simply uses a one-hot vector to represent different tasks. The result shown in Figure 9 demonstrates that different task prompts indeed affect the performance.

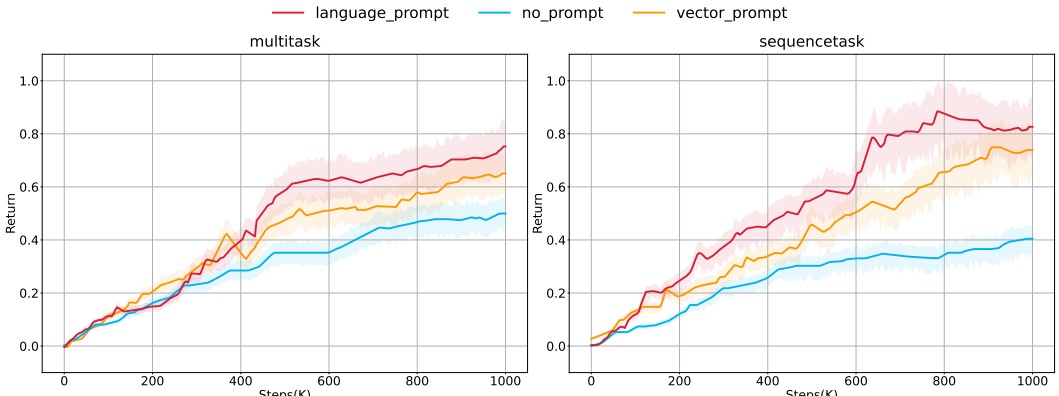

Figure 9: Ablation study on the design choices of task prompt. Different types of task prompts lead to varied behavior and performance.

**Effect of the random dropping strategy.** Here we show that the random dropping strategy is very important in the online adaptation process with the total reward indicator. In our online adaptation phase, when using the total reward importance indicator, the importance of any trajectory will never change during the whole training process. Hence, trajectories with high importance might always be used to update the generator, which inspired us to introduce a random dropping strategy. The ablation study conducted on the D4RL Locomotion Environments (Fu et al., 2020) shown in Figire 10 illustrates that the random dropping strategy can significantly improve the performance.

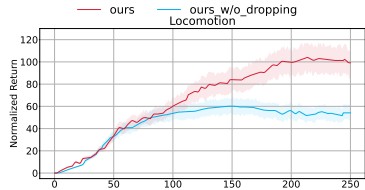

Figure 10: Ablation study on the random dropping strategy. The strategy can indeed improve the performance.

### C.3 FULL RESULTS

Here we present the full results for D4RL Locomotion with all 4 different offline datasets. Figure 11 shows that our ATraDiff is comparable to or better than the original RL baseline method with low-quality data (like random data), and the performance gap between our ATraDiff and the baseline becomes much more pronounced with medium/high-quality data (like medium-expert data).

### C.4 TRAINING TIME

The experiments are conducted on a single NVIDIA RTX 4090TI GPU. The training time of ATraDiff is listed in Table 4.

|  | Walker2D | Hopper | Halfcheetal |
|---|---|---|---|
| SAC | 6.5h | 7h | 7h |
| SAC with ATraDiff | 15h | 17h | 16.5h |
| RDEQ | 6h | 6.5h | 6.5h |
| RDEQ with ATraDiff | 14.5h | 16.5h | 16h |

Table 4: Time cost (hours) of training SAC and RDEQ with and without ATraDiff.

### C.5 ONLINE EXPERIMENTS FOR SYNTHER

We now include comparisons between our method and SynthER (Lu et al., 2023) in the online setting. We have conducted experiments on the Meta-World Environments. This experiment is

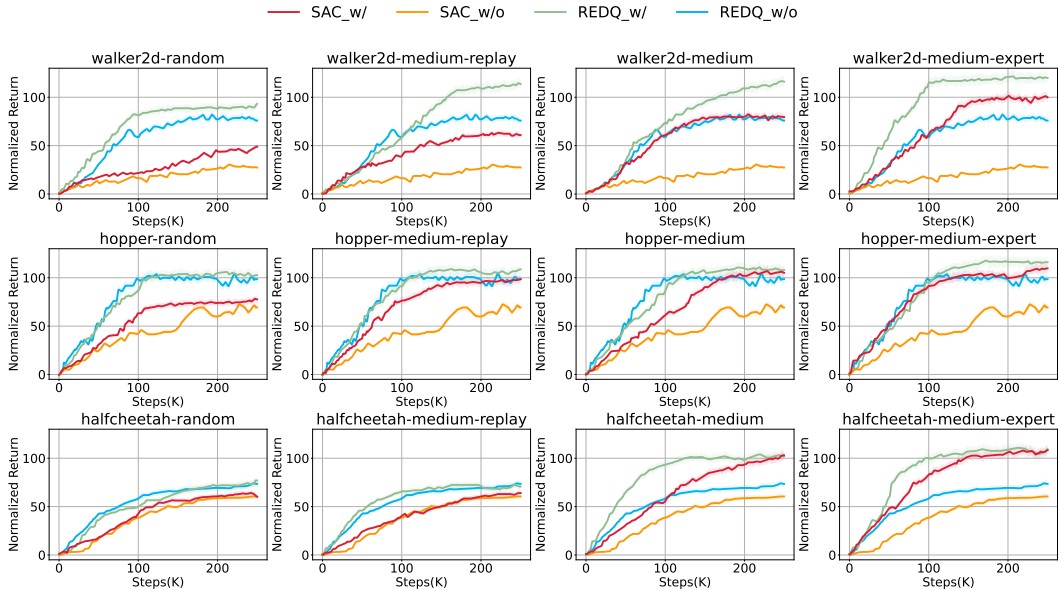

Figure 11: Learning curves on the D4RL Locomotion benchmark. ATraDiff (denoted as 'w/') is comparable to or better than the original RL baseline method with low-quality data, and the performance gap between ATraDiff and the baseline becomes much more pronounced with medium/high-quality data.

crucial to demonstrate the adaptability and effectiveness of our approach in dynamic and complex settings. For a fair comparison, we treat SynthER same as our method, combining it with RL methods using the method in Section 4.2. The results shown in Figure 12 indicate a significant performance advantage of our method over SynthER in these online environments.

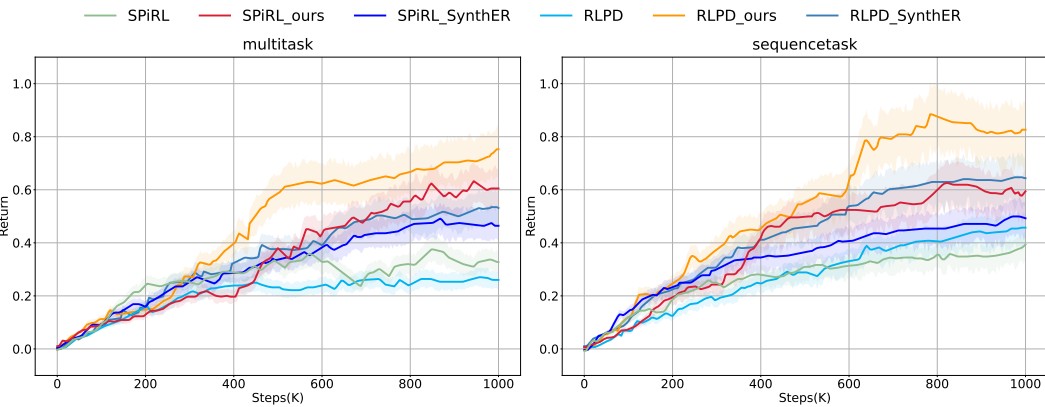

Figure 12: Experiment on the Meta-World benchmark. ATraDiff performs significantly better than SynthER on complicated environments.

## C.6   REWARD PREDICTOR UNDER ONLINE SETTINGS

To verify that the learned reward predictor in Section C.1 also works under the online reinforcement learning environments, we conduct experiments that compare SAC and SAC combined with our method with a learned reward predictor. The experiments are done in 3 D4RL environments, with the medium-expert datasets. The results in Figure 13 show that our method with the reward predictor also improves the performance of RL methods under the online setting.

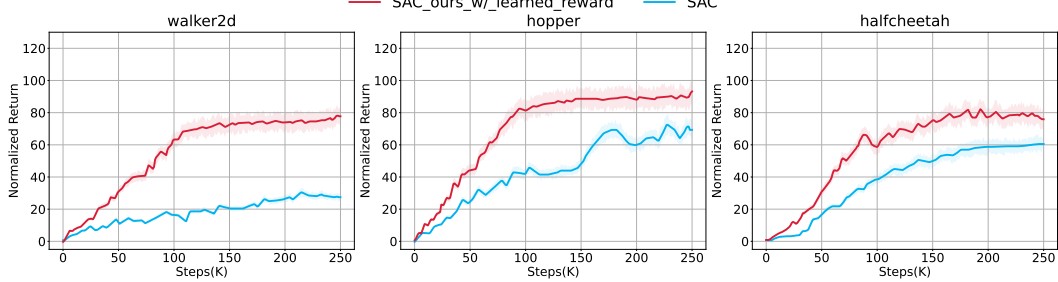

Figure 13: Experiments on the learned reward predictor in online reinforcement learning. Our method with the reward predictor still significantly improves the performance of online RL methods.

## D  VISUALIZATIONS

In this section, we show some representative visualizations of our generated image trajectories. We present a variety of images generated under different conditions, including varying initial states and a range of tasks. The labels in the left of the figures represent the task. The results demonstrate the robustness and effectiveness of our image generation method, consistently performing well across diverse scenarios and producing high-quality and temporally coherent image trajectories.

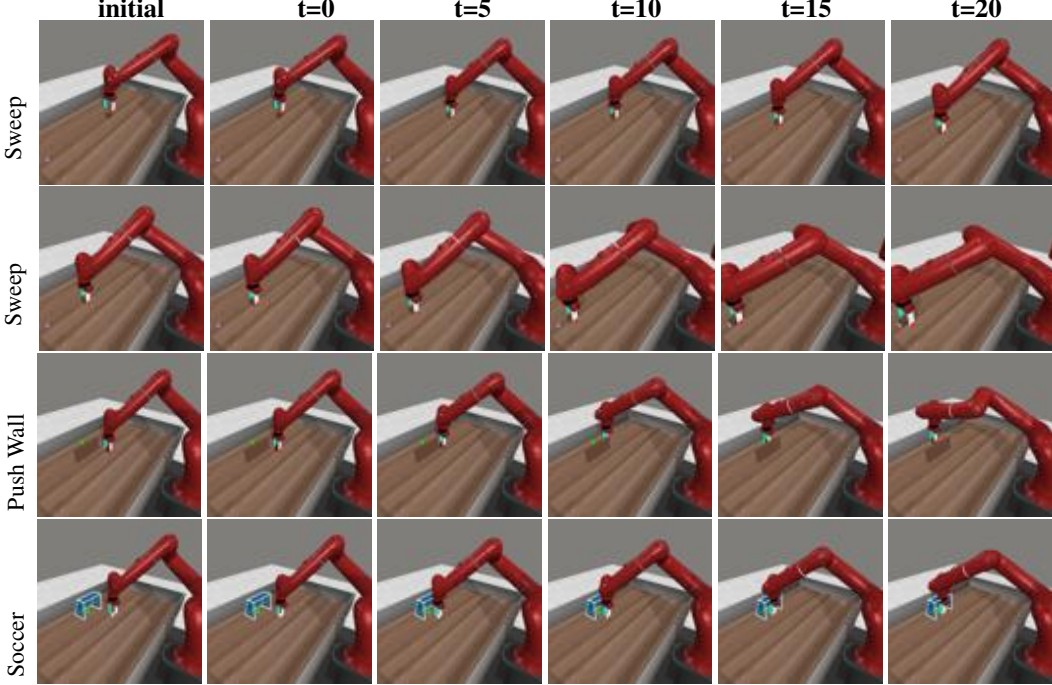

