# OpenReview forum: "ATraDiff: Accelerating Online Reinforcement Learning with Imaginary Trajectories"
_ICLR.cc/2024/Conference — Submitted to ICLR 2024_

### Official Review · Reviewer_rtc7 · 2023-10-22

**Soundness:** 2 fair
**Presentation:** 1 poor
**Contribution:** 3 good
**Rating:** 3
**Confidence:** 5

**Summary:**

The paper introduces a method to generate synthetic trajectories for online reinforcement learning using diffusion models, using offline data to bootstrap data generation. The training data for the diffusion model is generated by projecting low-dimensional states to images and then stacking them in a single image. The authors then fine-tune Stable Diffusion with this dataset, with the option to continuously update with online data. Generated images are then projected back to low-dimensional states. The resulting approach improves performance on a number of standard benchmarks.

**Strengths:**

- The problem setting is well-motivated and important to data efficient reinforcement learning.
- The paper proposes an interesting approach to generating trajectories in image-space by fine-tuning Stable Diffusion with stacked images.
- The approach shows promising results across a wide variety of datasets and benchmarks, with ablations on relevant portions of the algorithm.

**Weaknesses:**

While the paper shows promising improvements over the non-augmented baseline, unclear presentation and missing details make it hard to determine the contribution of the paper. In particular,
- **(Incomplete description of algorithm)** A huge amount of detail is missing when the authors claim they convert states to images and then back to states in Section 4.1. This is highly non-trivial and should be justified further. For example, can the authors present pictures of rendered states, what is the error from converting to image and back again? There is also no description of how Stable Diffusion was fine-tuned or how long online image generation takes. Furthermore, it is not explained how the authors would generate low-dimensional actions or obtain rewards from an image model.
- **(Benefit of trajectories vs. transitions)** The paper claims “full trajectories offer a more comprehensive source of information, enabling RL agents to better learn from past experiences” without evidence in the introduction. No comparison is made to the transition-based baseline in [1] which also evaluates on the same MuJoCo locomotion environments in Figure 2. Furthermore, [3] also proposes a trajectory based method for generating synthetic data with diffusion models and is not discussed.
- **(Why project states to images)** The paper claims that generating a collection of images outperforms single states in Figure 5. It is unclear why the simpler option of generating a sequence of states as in [2, 3] is not chosen. Given the image-based approach, it may also be asked why the authors do not focus on upsampling image datasets from the start, e.g. from [4].
- **(Incomplete experimental description)** Unclear what experimental setup is used in Figure 2. Is each figure the average of 4 different D4RL datasets? No explanation for what environments or datasets are used in Figure 5.
- **(Unclear description of sampling scheme)** There is an extremely vague description of what the reward or TD-error based pickup strategy means in Appendix A.1.1.

Minor
- The description of the state diffusion model in Appendix A.2 is copied nearly verbatim from Appendix B.2 of [1]. The authors should reword this to avoid plagiarism and cite this appropriately.


[1] Synthetic Experience Replay. Cong Lu, Philip J. Ball, Yee Whye Teh, and Jack Parker-Holder. NeurIPS, 2023.

[2] Planning with Diffusion for Flexible Behavior Synthesis. Michael Janner*, Yilun Du*, Joshua Tenenbaum, and Sergey Levine. ICML, 2022.

[3] Diffusion Model is an Effective Planner and Data Synthesizer for Multi-Task Reinforcement Learning. Haoran He, Chenjia Bai, Kang Xu, Zhuoran Yang, Weinan Zhang, Dong Wang, Bin Zhao, Xuelong Li. NeurIPS, 2023.

[4] Challenges and Opportunities in Offline Reinforcement Learning from Visual Observations; Cong Lu, Philip J. Ball, Tim G. J. Rudner, Jack Parker-Holder, Michael A. Osborne, Yee Whye Teh. TMLR, 2023.

**Questions:**

I would appreciate clarifications and responses to each of the concerns in the weaknesses section.

---

> ### Author Response · Authors · 2023-11-23
> **Response to Reviewer rtc7 (Part 1)**
>
> Thanks a lot for your insightful and inspiring comments! We provide the following clarifications in response to your concerns:
>
> 1. Incomplete description of algorithm
>
>     - Thank you for pointing out the missing details in our methodology. To address this, we have revised our manuscript thoroughly and included the previously omitted methodology and implementation details, including those pointed out by the reviewer (e.g., the conversion method between images and states) and other details (e.g., the architecture of the generation models and the pick-up strategy). These details are discussed in the updated main paper, and a detailed  explanation is available in Section A of the updated appendix. In addition, we will release our code upon acceptance. We hope that the revision improves the clarity and provides a thorough understanding of our method and procedure. For ease of reference, below we briefly summarize the missing details raised by the reviewer, with more in-depth information available in the revision.
>
>     - **(converting between images and states)** For the renderer converting states to images, we follow the previous work [Ref1]. This involves running the agent in the environment and capturing observations as rendering results. The recognizer, which converts images back to states, comprises a general convolutional component and separate MLPs. These MLPs are tasked with estimating the 3D positions of key points, tailored to different environments.
>
>     - **(Pictures of rendered states)** In Appendix D of the updated appendix, we have included representative visualizations of our generated image trajectories. We present a variety of images generated under different conditions, including varying initial states and a range of tasks. The results demonstrate the robustness and effectiveness of our image generation method, consistently performing well across diverse scenarios and producing high-quality and temporally coherent image trajectories.
>
>     - **(Conversion error between images and states)** 1) The error of converting states to images and then back to states is relatively small. For example, during the training of our recognizer in Meta-World, we found that the loss, which represents the error, is less than $0.054$. 2) Important, we would like to point out that in our case, we are using the synthesized data to train RL methods, not for planning. This means that we do not require the conversion to be very accurate. Instated, we can tolerate the errors within this process and even benefit increased robustness from this.
>
>     - **(Description of the generation model and fine-tuning Stable Diffusion)** We have set the generation lengths for multiple generators at 5, 10, 15, 20, and 25, respectively. For fine-tuning each generator corresponding to length $k$, we use the offline dataset to generate training data, creating sub-trajectories from points in full trajectories. Sub-trajectories shorter than length $k$ are padded to length $k$ using the last frame. We use LoRA to fine-tune the Stable Diffusion based generators. Additionally, we have conducted an experiment on the wall-clock running time of our generator, detailed in Section C.4 of the updated appendix. We would like to note that in the context of online reinforcement learning, the most significant cost often lies in interacting with the real environment and data collection. Consequently, our primary focus with ATraDiff has been on optimizing sample efficiency to minimize the time required for environmental interaction.
>
>     - **(Action and reward generation)** We would like to clarify that the action and reward are not generated directly from images. Instead, they are derived from states obtained via the recognizer. More concretely, actions are calculated based on adjacent states, specific to the task, while rewards are generated through interaction with the ground-truth environment. Note that in the context of online training, RL methods inherently involve accessing the environment. Additionally, in the updated manuscript, we have also included a variant of our method that trains a reward prediction function from an offline dataset. The effectiveness of this reward predictor has been demonstrated in our offline experiments, the results of which are included in Section C.1 of the updated appendix.

---

> ### Author Response · Authors · 2023-11-23
> **Response to Reviewer rtc7 (Part 2)**
>
> 2. Benefits of trajectories vs. transitions: Comparison with SynthER
>
>     - First, we would like to clarify that **the task settings differ between our work and SynthER**. Our work primarily focuses on leveraging offline data to enhance online reinforcement learning performance. In contrast, SynthER [Ref2] aims to upsample the offline dataset using their data synthesizer. A crucial distinction lies in their methodology for online experiments, where they do not integrate the offline dataset, which is a key component in our approach. This fundamental difference in task settings initially led us to exclude a direct comparison with SynthER.
>
>     - Additionally, at the time of our submission to ICLR, SynthER was not yet published, appearing later in NeurIPS 2023. Given these circumstances, a direct comparison was not feasible in our initial submission.
>
>     - Following the reviewer’s suggestion, we now include comparisons between our method and SynthER. Consistent with the discussion above, we consider two types of comparisons: **(I) Employ SynthER in our task setting, which we care most about**. To this end, we have conducted a comparative analysis in an **online** setting within the Meta-World Environments. This analysis is crucial to demonstrate the adaptability and effectiveness of our approach in dynamic and complex settings. **Our results indicate a significant performance advantage of our method over SynthER in these online environments**, as summarized in the table below. For more detailed results, including full performance curves, please refer to Section C.5 of the updated appendix.
>
>     |Task Name|SPiRL+SynthER|SPiRL+ours|RLPD+SynthER|RLPD+ours|
>     |---|:---:|:---:|:---:|:---:|
>     |multitask|0.45|0.61|0.52|0.75|
>     |sequencetask|0.49|0.57|0.62|0.80|
>
>     - **(II) Employ our method in SynthER’s task setting, which is not the focus of our paper**. To this end, we have included an additional comparison in Section C.1 of the updated appendix. This comparison, conducted in the D4RL Locomotion environment under **offline** settings, demonstrates that our method can achieve comparable or superior results to SynthER.
>
>     |Task Name|TD3+BC|TD3+BC+SynthER|TD3+BC+ours|IQL|IQL+SynthER|IQL+ours|
>     |---|:---:|:---:|:---:|:---:|:---:|:---:|
>     | halfcheetah-random | 11.3 | 12.2 | 12.5 | 15.2 | 17.2 | 17.1|
>     | halfcheetah-medium | 48.1 | 49.9 | 52.3 | 48.3 | 49.6 | 53.1|
>     | halfcheetah-replay | 44.8 | 45.9 | 46.5 | 43.5 | 46.7 | 49.2 |
>     | halfcheetah-expert | 90.8 | 87.2 | 93.6 | 94.6 | 93.3 | 95.2 |
>     | hopper-random | 8.6 | 14.6 | 15.2 | 7.2 | 7.7 | 8.1|
>     | hopper-medium | 60.4 | 63.4 | 65.7 | 62.8 | 72.0 | 72.4 |
>     | hopper-replay | 64.4 | 53.4 | 64.7 | 84.6 | 103.2 | 103.6 |
>     | hopper-expert | 101.1 | 105.4 | 111.2 | 106.2 | 90.8 | 113.6 |
>     | walker-random | 0.6 | 2.3 | 2.1 | 4.1 | 4.2 | 4.3|
>     | walker-medium | 82.7 | 84.8 | 87.5 | 84.0 | 84.7 | 89.1 |
>     | walker-replay | 85.6 | 90.5 | 86.3 | 82.6 | 83.3 | 85.4 |
>     | walker-expert | 110.0 | 110.2 | 111.2 |111.7 |111.4 | 111.7 |
>
>     - In summary, our aforementioned experimental results validate that, **in both our task setting and SynthER’s task setting, our trajectory-level generation outperforms the transition-level generation of SynthER**.
>
> 3. Benefits of trajectories vs. transitions: Difference from [Ref3]
>
>     - Our work is different from [Ref3] in two important ways:  1) [Ref3] actually addresses a task different from our work. They focus on the planning phase, while we focus on the online training aspect. 2) Different from the one-time generation in their work, we leverage multiple generations from different start points with different tasks which deal with the unique challenges associated with flexible generation and distribution shift. We have discussed these differences in the Related Works section.

---

> ### Author Response · Authors · 2023-11-23
> **Response to Reviewer rtc7 (Part 3)**
>
> 4. Why projecting states into images
>
>     - We would like to emphasize that **our work develops a general plug-in data generation method to facilitate online RL algorithms, agnostic to the specific form of generation, which could either be pixel/image-level based or state-level based**. For example, the important deployment method of our generator and online adaptation technique are agnostic to the form of generation. In our paper, we have investigated the two variants of our data generation method: at image or state levels, as shown in Fig. 6. And we found that 1) both image and state generation variants significantly outperform the baselines; 2) image generation works better under our paradigm. Our hypothesis for the second finding is that image generation can benefit from a pre-trained generation model (e.g., Stable Diffusion) while state generation is trained from scratch, though image generation needs additional modules to convert between images and states.
>
>     - In principle, our method could still be applicable to those vision-based reinforcement learning tasks as suggested by the reviewer, not only restricted to the benchmarks we investigated in the paper. Following previous work [Ref2], this paper mainly focuses on traditional offline RL datasets like D4RL. Evaluating these vision-based tasks is an interesting direction for future work.
>
> 5. Incomplete experimental descriptions
>
>     - For Fig. 3 (formerly Fig. 2 in the reviewer’s original comment – we included another figure in the updated manuscript), in the original manuscript, we followed prior studies [Ref4] and presented the average performance across different datasets from the D4RL benchmark. To provide a more comprehensive understanding, we have now included separate results for datasets in Section C.3 of the updated appendix.
>
>     - In Fig. 6 (formerly Fig. 5 in the reviewer’s original comment – we included another figure in the updated manuscript), we conducted experiments on the D4RL Locomotion environments. We have included the missing description to the caption in the updated manuscript.
>
> 6. Unclear description of sampling scheme
>
>     - We apologize for any confusion. We have thoroughly revised Section A.1 of the updated appendix, providing detailed explanation on the pick-up strategy. We also elaborate on the various indicators used and how they influence sample selection in our model. Below we briefly summarize the key points:
>
>     - We investigated different indicators to represent the importance of the real samples collected from interacting with the environment when running online RL methods. These indicators are designed to discern different properties, influencing which samples are considered more crucial for effective learning and adaptation.
>
>     - To ensure that the selected samples used to adapt the generator are suitable, we developed 'pick-up strategies' according to different indicators. These strategies involve a systematic approach to maintaining a data structure that stores real samples and selecting the most appropriate ones for online adaptation. The intention is to ensure that our model utilizes the most valuable samples, as indicated by our chosen metrics, to optimize learning outcomes.
>
>     - The *TD-error indicator* uses the TD-error of a transition to represent its importance. It has the property of flexible importance during the training process. Therefore, we use a large buffer to store more samples, and select samples with the most importance each time when we need to update. In contrast, the *reward indicator* uses the total reward of a trajectory to represent its importance, which has the property of fixed importance. To avoid using fixed samples to update the generator too many times, we have designed a small buffer with a random dropping strategy.
>
> 7. Description in Appendix A.2
>
>     - We apologize for this and appreciate the reviewer for pointing it out. We have rewritten this section as well as improved its clarity.
>
>     - We hope that our clarifications have addressed the reviewer’s concern. We will also release our code upon acceptance. If there are any remaining details or concerns that require further clarification, please do not hesitate to let us know. We are committed to ensuring that our research is transparent and accessible.
>
> References:
>
> [Ref1] Younggyo Seo, Danijar Hafner, Hao Liu, Fangchen Liu, Stephen James, Kimin Lee, and Pieter Abbeel. Masked world models for visual control. CoRL 2022.
>
> [Ref2] Cong Lu, Philip J. Ball, Yee Whye Teh, and Jack Parker-Holder. Synthetic Experience Replay. NeurIPS, 2023.
>
> [Ref3] Haoran He, Chenjia Bai, Kang Xu, Zhuoran Yang, Weinan Zhang, Dong Wang, Bin Zhao, Xuelong Li. Diffusion Model is an Effective Planner and Data Synthesizer for Multi-Task Reinforcement Learning. NeurIPS, 2023.
>
> [Ref4] Philip J. Ball, Laura M. Smith, Ilya Kostrikov, and Sergey Levine. Efficient online reinforcement learning with offline data. ICML 2023.

---

> > ### Comment · Reviewer_rtc7 · 2023-12-01
> > **Thank you for the response.**
> >
> > I deeply appreciate the author's efforts in providing clarity to the algorithm they propose and providing additional baselines. As has been discussed elsewhere in the responses, the author's clarifications paint a very different picture to what was previously presented. In particular, choices like using the ground truth reward make the method far more restrictive than before.
> >
> > These requirements together with the choice to project to images and then back to states beg the question why the simpler approach to generate low-dimensional sequences of transitions was not chosen as in [Ref3]. I believe the authors may have missed that in [Ref3] they also show experiments training on the synthetic trajectories, rather than just planning.

---

### Official Review · Reviewer_8Lyo · 2023-11-01

**Soundness:** 3 good
**Presentation:** 3 good
**Contribution:** 2 fair
**Rating:** 5
**Confidence:** 3

**Summary:**

The paper introduces ATraDiff, which uses diffusion models to generate synthetic trajectories for online RL. It can augment the replay buffer of any online RL algorithm with full trajectories, improving data efficiency and performance. ATraDiff can handle varying trajectory lengths and distribution shifts between offline and online data. It achieves state-of-the-art results on several benchmarks, particularly in complex environments with sparse rewards.

**Strengths:**

- It proposes a novel method that uses diffusion models to generate synthetic trajectories for online RL, which is a challenging and important problem due to distributional shift.
- It can improve the data efficiency and performance of any online RL algorithm as a plug-in method, by augmenting the replay buffer with full trajectories that are conditioned on the current state and task.
- It can handle varying trajectory lengths and distribution shifts between offline and online data by using a coarse-to-precise strategy and an online adaptation mechanism.

**Weaknesses:**

See questions.

**Questions:**

1. In Sec. 4.2, is the synthetic trajectory actually $(\ldots,s_L,a_L)$ instead of $(\ldots, s_t,a_t)$?
2. How can the reward $R(s_i,a_i)$ of a synthetic state-action pair be calculated from the environment if the action is not executed in the environment, or is the reward function accessible to the agent?
3. In the last part of Sec. 4.3, it states that "Hence, we will randomly drop some samples from the maintained subset regardless of
their importance". Is there any ablation study to support the effectiveness of this random dropping strategy?
4. A recent work [1] about offline and online data augmentation using diffusion models has been published. Could you please include some comparison between your work and [1]? Experimental comparison will be better.

[1] Synthetic Experience Replay. NeurIPS 2023. https://arxiv.org/abs/2303.06614

---

> ### Author Response · Authors · 2023-11-23
> **Response to Reviewer 8Lyo (Part 1)**
>
> Thanks a lot for your insightful and inspiring comments! We provide the following clarifications in response to your concerns:
>
> 1. Technique detail in Section 4.2
>
>     - In Sec. 4.2, the synthesized result is actually $(\dots, s_t, a_t)$, not $(\dots, s_L, a_L)$. Here $L$ represents the expected length of the generated trajectories, which is not the real length. In practice, the generated length can be flexible, due to the multiple generators and trajectory pruner.
>
> 2. Reward generation
>
>     - The reward is calculated by accessing the ground truth environment. It is important to note that **in the context of online training, RL methods inherently involve accessing the environment**. Our work focuses on the paradigm that utilizes the offline data to improve the performance of online reinforcement learning, aiming to develop a general tool that can be applied to any online reinforcement methods. Therefore, our pipeline involves accessing the online environment in the online training phase, which might be different from traditional offline learning methods.
>
>     - We would like to also highlight that our ATraDiff's interaction with the environment is deliberately constrained in our method. As detailed in Section 4.2, the environment access probability is set as $\frac{\rho}{(1-\rho)L}$. Here $L$ represents the expectation length of generation, $T$ represents the real sample time, and $\rho$ is the controlling variable. This ensures that environment interactions with ATraDiff occur only $\frac{\rho} * T$ times, a constant and limited number, thereby controlling the potential increase in time costs.
>
>     - In the updated manuscript, we have also included a variant of our method that trains a reward prediction function from an offline dataset. To achieve this, we have followed the established methodology from prior work [Ref1], and implemented supervised training with the offline dataset. The effectiveness of this reward predictor has been demonstrated in our offline experiments, the results of which are included in Section C.1 of the updated appendix as well as shown in the table below. In our future work, we aim to adapt this reward predictor for online scenarios and further improve the performance.
>
>     |Task Name|TD3+BC|TD3+BC+ours_with_learned_reward|IQL|IQL+ours_with_learned_reward|
>     |---|:---:|:---:|:---:|:---:|
>     |halfcheetah-random|11.3 | 11.7 | 15.2 | 16.8|
>     |halfcheetah-medium|48.1 | 49.3 | 48.3 | 50.2|
>     |halfcheetah-medium-replay|44.8 | 45.3 | 43.5 | 45.6|
>     |halfcheetah-medium-expert|90.8 | 91.9 | 94.6 | 94.5|
>     |hopper-random|8.6 | 14.8 | 7.2 | 8.3 |
>     |hopper-medium|60.4 | 64.5 | 62.8 | 72.2|
>     |hopper-medium-replay| 64.4 | 65.2 | 84.6 | 97.0|
>     |hopper-medium-expert| 101.1 | 108.7 | 106.2 | 108.3 |
>     |walker-random| 0.6 | 3.4 | 4.1 | 4.0 |
>     |walker-medium|82.7 | 85.0 | 84.0 | 85.3 |
>     |walker-medium-replay|85.6 | 87.5 | 82.6 | 84.9|
>     |walker-medium-expert|110.0 | 111.1 | 111.7 | 111.6 |
>
> 3. Random dropping strategy
>
>     - In the online adaptation phase of our pipeline, we employed the total reward importance indicator to assess the significance of each trajectory. An inherent characteristic of this method is that the importance assigned to any given trajectory remains constant throughout the training process. Consequently, trajectories with high importance are perpetually prioritized for generator updates.
>
>     - This consistent prioritization led us to implement a *random dropping* strategy. By introducing this strategy, we aimed to inject variability into the training process, mitigating potential biases that might arise from the continuous use of high-importance trajectories. We believe that this approach not only diversifies the training data but also enhances the robustness of our generator.
>
>     - To evaluate the effectiveness of the random dropping strategy, we have conducted an ablation study. The result and discussion are detailed in Section C.2 of the updated appendix, showing that the random dropping strategy significantly improves the performance.

---

> > ### Comment · Reviewer_8Lyo · 2023-11-23
> >
> > Thanks for your detailed reply and additional experiments. The experiment results show that ATraDiff performs well under various settings and environments. However, I cannot agree with your explaination about accessing the ground truth reward function. Though RL agents can interact with the online environment, it is not clear that they can directly access the reward function. Online RL means that agents can receive the feedback of their actions from the environment, which they cannot in offline RL. If the ground truth reward function is defaultly accessible, it is redundant for many MBRL algorithms to learn an additional reward model besides the dynamics model.
> >
> > It seems that your supplementary experiments with the learned reward function also performs well. I think the paper will be more solid if your take the similar scheme in the online RL part.

---

> > > ### Author Response · Authors · 2023-11-23
> > > **Response to Reviewer 8Lyo**
> > >
> > > We thank the reviewer for the reply and follow-up. We are glad to hear that our response has addressed most of the reviewer’s concerns. Below, we would like to address the remaining concern on accessing the ground-truth reward.
> > >
> > > - First, we would like to point out the difference between our method and regular MBRL methods. **In our method, the reward is not generated by directly calling some known reward function $R(s,a)$. Instead, the reward is obtained by interacting the agent with the environment**, which could be more reasonable than directly having the reward function.
> > >
> > > - Second, **our method still achieves superior performance with the learned reward function in online reinforcement learning environments, as shown by the experimental results in the newly added Section C.6 of the updated appendix**. Due to the time and computational resource constraints during the rebuttal period, so far we have generated the full suite of offline learning results and a portion of online learning results with the learned reward function. We believe that these results have already demonstrated the effectiveness of our method with the learned reward function. And we are currently working on the remaining online learning results and will include them in the next revision.
> > >
> > > - Finally, we would like to reiterate that this paper is focused on developing a general plug-in data generation method to facilitate online RL algorithms, rather than proposing a new reward function learning algorithm. We have empirically shown that our method works with both ground-truth reward obtained by interacting the agent with the environment and the reward function learned by existing method, both of which significantly improve the online RL algorithms.
> > >
> > > We hope that these clarify the review’s remaining concern. We are more than happy to discuss, if the reviewer has any further questions.

---

> > > > ### Comment · Reviewer_8Lyo · 2023-11-23
> > > >
> > > > Thanks for your further reply. I am glad to admit that the performance of ATraDiff with learned reward functions is also excellent. But I am still curious about how the reward $R(s,a)$ can be obtained by interacting with the environment, unless the agent is set to state $s$ and takes action $a$. In Section 4.2, the synthetic transitions $(s_i,a_i)$ are not actually executed from my personal perspective. If my understanding is wrong, I am willing to hear the correct scheme of your algorithm.

---

> ### Author Response · Authors · 2023-11-23
> **Response to Reviewer 8Lyo (Part 2)**
>
> 4. Lack of comparison with SynthER
>
>     - First, we would like to clarify that **the task settings differ between our work and SynthER**. Our work primarily focuses on leveraging offline data to enhance online reinforcement learning performance. In contrast, SynthER aims to upsample the offline dataset using their data synthesizer. A crucial distinction lies in their methodology for online experiments, where they do not integrate the offline dataset, which is a key component in our approach. This fundamental difference in task settings initially led us to exclude a direct comparison with SynthER.
>
>     - Additionally, at the time of our submission to ICLR, SynthER was not yet published, appearing later in NeurIPS 2023. Given these circumstances, a direct comparison was not feasible in our initial submission.
>
>     - Following the reviewer’s suggestion, we now include comparisons between our method and SynthER. Consistent with the discussion above, we consider two types of comparisons: **(I) Employ SynthER in our task setting, which we care most about**. To this end, we have conducted a comparative analysis in an **online** setting within the Meta-World Environments. This analysis is crucial to demonstrate the adaptability and effectiveness of our approach in dynamic and complex settings. **Our results indicate a significant performance advantage of our method over SynthER in these online environments**, as summarized in the table below. For more detailed results, including full performance curves, please refer to Section C.5 of the updated appendix.
>
>     |Task Name|SPiRL+SynthER|SPiRL+ours|RLPD+SynthER|RLPD+ours|
>     |---|:---:|:---:|:---:|:---:|
>     |multitask|0.45|0.61|0.52|0.75|
>     |sequencetask|0.49|0.57|0.62|0.80|
>
>     - **(II) Employ our method in SynthER’s task setting, which is not the focus of our paper**. To this end, we have included an additional comparison in Section C.1 of the updated appendix. This comparison, conducted in the D4RL Locomotion environment under **offline** settings, demonstrates that our method can achieve comparable or superior results to SynthER.
>
>     |Task Name|TD3+BC|TD3+BC+SynthER|TD3+BC+ours|IQL|IQL+SynthER|IQL+ours|
>     |---|:---:|:---:|:---:|:---:|:---:|:---:|
>     | halfcheetah-random | 11.3 | 12.2 | 12.5 | 15.2 | 17.2 | 17.1|
>     | halfcheetah-medium | 48.1 | 49.9 | 52.3 | 48.3 | 49.6 | 53.1|
>     | halfcheetah-replay | 44.8 | 45.9 | 46.5 | 43.5 | 46.7 | 49.2 |
>     | halfcheetah-expert | 90.8 | 87.2 | 93.6 | 94.6 | 93.3 | 95.2 |
>     | hopper-random | 8.6 | 14.6 | 15.2 | 7.2 | 7.7 | 8.1|
>     | hopper-medium | 60.4 | 63.4 | 65.7 | 62.8 | 72.0 | 72.4 |
>     | hopper-replay | 64.4 | 53.4 | 64.7 | 84.6 | 103.2 | 103.6 |
>     | hopper-expert | 101.1 | 105.4 | 111.2 | 106.2 | 90.8 | 113.6 |
>     | walker-random | 0.6 | 2.3 | 2.1 | 4.1 | 4.2 | 4.3|
>     | walker-medium | 82.7 | 84.8 | 87.5 | 84.0 | 84.7 | 89.1 |
>     | walker-replay | 85.6 | 90.5 | 86.3 | 82.6 | 83.3 | 85.4 |
>     | walker-expert | 110.0 | 110.2 | 111.2 |111.7 |111.4 | 111.7 |
>
>
>     - In summary, our aforementioned experimental results validate that, **in both our task setting and SynthER’s task setting, our trajectory-level generation outperforms the transition-level generation of SynthER**.
>
>
> References:
>
> [Ref1] Ksenia Konyushkova, Konrad Zolna, Yusuf Aytar, Alexander Novikov, Scott Reed, Serkan Cabi, and Nando de Freitas. Semi-supervised reward learning for offline reinforcement learning. NeurIPS Workshop, 2020.
>
> [Ref2] Cong Lu, Philip J. Ball, Yee Whye Teh, and Jack Parker-Holder. Synthetic Experience Replay. NeurIPS, 2023.

---

### Official Review · Reviewer_cnq2 · 2023-11-04

**Soundness:** 3 good
**Presentation:** 3 good
**Contribution:** 3 good
**Rating:** 6
**Confidence:** 4

**Summary:**

The paper proposes ATraDiff, a data augmentation method to generate synthetic trajectories with diffusion models to enhance online RL and offline-to-online RL. The method is able to handle varying trajectory lengths, and overcome offline-to-online data distribution shifts. Orthogonally applicable to RL methods with a replay buffer, ATraDiff improves their performance on several benchmarks, particularly in complicated tasks.

**Strengths:**

- Good performance improvement, orthogonally applicable to any RL algorithm with a replay buffer.

**Weaknesses:**

- The additional latency of using ATraDiff, and the continual training of the diffusion model on new experiences is not mentioned, but it's a rather important detail.
- Figure line names ordering and colours are not very intuitive to follow
- Would have liked to see an analysis on the quality of the images generated by ATraDiff, particularly it would be interesting to see how temporally correlated are the trajectories.
- Overall the paper would benefit from more in depth analysis of the ablation studies too. Would like to see comparison with the state-based diffuser on more tasks, as well as a full comparison with it's closest method SynthER, both in terms of performance and computational requirements.
- Unsure how reproducible the paper is as many implementation details are missing.

**Questions:**

- How temporally correlated are the image trajectories generated by diffusion? Does it create a coherent video? A good analysis on this would definitely strengthen the paper.
- What is the added latency of using ATraDiff compared to only using one of the baseline RL algorithms? Regardless if it's significantly more computationally expensive, it would still be helpful to know quantitatively.
- Not sure I understand why there is no comparison with its closest method SynthER?

To recap, a more insightful analysis on the kind and quality of trajectories generated, and a comparison with SynthER could help improve the score.

Edit: Rating improved after concerns were addressed by the authors response.

---

> ### Author Response · Authors · 2023-11-23
> **Response to Reviewer cnq2 (Part 1)**
>
> Thanks a lot for your insightful and inspiring comments! We provide the following clarifications in response to your concerns:
>
> 1. Additional latency of using ATraDiff
>     - Our ATraDiff is designed with a trade-off between sample efficiency and computational resource demands. We would like to emphasize that **in the context of online reinforcement learning, the most significant cost often lies in interacting with the real environment and data collection**. Consequently, our primary focus with ATraDiff has been on optimizing sample efficiency to minimize the time required for environment interaction.
>     - Following the reviewer’s suggestion, we quantify the time impact of ATraDiff, by recording the total training time during experiments on the D4RL Locomotion dataset, running on a single NVIDIA RTX 4090TI GPU. The results are shown in the table below, which are also included in Table 4 of the updated Appendix. The comparison between SAC/REDQ and SAC/REDQ with ATraDiff shows that integrating ATraDiff with RL methods leads to an extension of the training duration. However, *this is counterbalanced by the improved efficiency in data utilization*.
>
>     ||walker2d|hopper|halfcheetah|
>     |---|---|---|---
>     |SAC|6.5h|7h|7h|
>     |SAC_with_ATraDiff|15h|17h|16.5h|
>     |REDQ|6h|6.5h|6.5h|
>     |REDQ_with_ATraDiff|14.5h|16.5h|16h|
>
>     - Moving forward, we are committed to enhancing the time efficiency of ATraDiff. Our goal is to reduce both training and inference times, while maintaining its current performance levels. We appreciate the reviewer’s valuable feedback and view it as a guiding direction for our future work.
>
> 2. Line names ordering and color in figures
>
>     - Thanks for your valuable feedback. We have adjusted the line name ordering and color in Figures 3, 4, and 5 of the updated manuscript.

---

> ### Author Response · Authors · 2023-11-23
> **Response to Reviewer cnq2 (Part 2)**
>
> 3. Lack of comparison with SynthER
>     - First, we would like to clarify that **the task settings differ between our work and SynthER**. Our work primarily focuses on leveraging offline data to enhance online reinforcement learning performance. In contrast, SynthER aims to upsample the offline dataset using their data synthesizer. A crucial distinction lies in their methodology for online experiments, where they do not integrate the offline dataset, which is a key component in our approach. This fundamental difference in task settings initially led us to exclude a direct comparison with SynthER.
>
>     - Additionally, at the time of our submission to ICLR, SynthER was not yet published, appearing later in NeurIPS 2023. Given these circumstances, a direct comparison was not feasible in our initial submission.
>
>     - Following the reviewer’s suggestion, we now include comparisons between our method and SynthER. Consistent with the discussion above, we consider two types of comparisons: **(I) Employ SynthER in our task setting, which we care most about**. To this end, we have conducted a comparative analysis in an **online** setting within the Meta-World Environments. This analysis is crucial to demonstrate the adaptability and effectiveness of our approach in dynamic and complex settings. **Our results indicate a significant performance advantage of our method over SynthER in these online environments**, as summarized in the table below. For more detailed results, including full performance curves, please refer to Section C.5 of the updated appendix.
>
>     |Task Name|SPiRL+SynthER|SPiRL+ours|RLPD+SynthER|RLPD+ours|
>     |---|:---:|:---:|:---:|:---:|
>     |multitask|0.45|0.61|0.52|0.75|
>     |sequencetask|0.49|0.57|0.62|0.80|
>     - **(II) Employ our method in SynthER’s task setting, which is not the focus of our paper**. To this end, we have included an additional comparison in Section C.1 of the updated appendix. This comparison, conducted in the D4RL Locomotion environment under **offline** settings, demonstrates that our method can achieve comparable or superior results to SynthER.
>
>     |Task Name|TD3+BC|TD3+BC+SynthER|TD3+BC+ours|IQL|IQL+SynthER|IQL+ours|
>     |---|:---:|:---:|:---:|:---:|:---:|:---:|
>     | halfcheetah-random | 11.3 | 12.2 | 12.5 | 15.2 | 17.2 | 17.1|
>     | halfcheetah-medium | 48.1 | 49.9 | 52.3 | 48.3 | 49.6 | 53.1|
>     | halfcheetah-replay | 44.8 | 45.9 | 46.5 | 43.5 | 46.7 | 49.2 |
>     | halfcheetah-expert | 90.8 | 87.2 | 93.6 | 94.6 | 93.3 | 95.2 |
>     | hopper-random | 8.6 | 14.6 | 15.2 | 7.2 | 7.7 | 8.1|
>     | hopper-medium | 60.4 | 63.4 | 65.7 | 62.8 | 72.0 | 72.4 |
>     | hopper-replay | 64.4 | 53.4 | 64.7 | 84.6 | 103.2 | 103.6 |
>     | hopper-expert | 101.1 | 105.4 | 111.2 | 106.2 | 90.8 | 113.6 |
>     | walker-random | 0.6 | 2.3 | 2.1 | 4.1 | 4.2 | 4.3|
>     | walker-medium | 82.7 | 84.8 | 87.5 | 84.0 | 84.7 | 89.1 |
>     | walker-replay | 85.6 | 90.5 | 86.3 | 82.6 | 83.3 | 85.4 |
>     | walker-expert | 110.0 | 110.2 | 111.2 |111.7 |111.4 | 111.7 |
>
>
>     - In summary, our aforementioned experimental results validate that, **in both our task setting and SynthER’s task setting, our trajectory-level generation outperforms the transition-level generation of SynthER**.
>
> 4. Missing analysis on the quality of generated images
>
>     - Thanks for your valuable suggestion. In Section D of the updated appendix, we have included representative visualizations of our generated image trajectories. We present a variety of images generated under different conditions, including varying initial states and a range of tasks. The results demonstrate the robustness and effectiveness of our image generation method, consistently performing well across diverse scenarios and producing high-quality and temporally coherent image trajectories.
>
> 5. Missing implementation detail and reproducibility
>
>     - Thank you for pointing out the missing implementation details. To address this, we have revised our manuscript and included the previously omitted methodology and implementation details, such as the architecture of the generation models, the conversion between images and states, and the pick-up strategy in Section A of the updated appendix. We hope that the revision improves the clarity and provides a thorough understanding of our method and procedure.
>
>     - Importantly, we will release our code upon acceptance to ensure reproducibility.
>
>     - If there are any remaining implementation details that require further clarification, please do not hesitate to let us know.

---

> ### Comment · Reviewer_cnq2 · 2023-11-23
>
> Thanks for addressing my concerns and answering the questions. I am satisfied by the answers and the updated paper, which I believe to be now a stronger submission, I have thus increased my rating.
>
> One more suggestion that I have for appendix D of the generated images, would be to show them side by side with real trajectories from the simulator, so that people that are not very familiar with the task can compare them more easily.

---

### Official Review · Reviewer_GZoQ · 2023-11-08

**Soundness:** 2 fair
**Presentation:** 3 good
**Contribution:** 2 fair
**Rating:** 3
**Confidence:** 3

**Summary:**

This work trained diffusion model on offline data to generate more training trajectories so that it can accelerate online RL learning.

The authors conducted experiments on two settings 1) online RL setting (3 envs) and 2) offline-to-online (hybrid) setting (3 envs). The experimental results shows improvement comparing to baseline without using generated trajectories.

**Strengths:**

- Using diffusion model to generate trajectories as data augmentation approach is novel, as far as I know.
- The approach is relatively straightforward and can be combined with a wide family of RL approaches.

**Weaknesses:**

- This approach assume the access to the ground truth reward function, which is a very impractical setting. Most of tasks evaluation of trajectories is expensive.
- This work lack of discussion on the assumptions of offline datasets that is already available to the approach. Is it a medium quality dataset? Is it a low-quality dataset? Does it contains trajectories from optimal policy?  My understanding is that this will make substantial difference to the algorithm performance. The D4RL dataset contains different level of offline data. It would be good to clarify the setting of offline datasets.
- In the online setting, the fair comparison would be using the offline data with the online algorithms. e.g., learning from demonstration approaches would be a better baseline. O.w. the proposed approach has an advantage of using more offline data. It's unclear to me whether the improvement comes from the access to the additional offline dataset or the proposed approach.

**Questions:**

- Did the author consider replacing the query. of reward function from env by training a reward function from offline trajectories?
Please also refer to the weakness section.

---

> ### Author Response · Authors · 2023-11-23
> **Response to Reviewer GZoQ (Part 1)**
>
> Thanks a lot for your insightful and inspiring comments! We provide the following clarifications in response to your concerns:
>
> 1. Having access to the ground truth reward
>
>     - We understand the reviewer’s concern on the potential costs associated with environment interaction and access to ground truth rewards. However, we would like to emphasize that **in the context of online training, RL methods inherently involve accessing the environment**, which aligns with the operational framework of our method. The core contribution of our work is the development of a versatile plug-in method, designed for integration with various online RL techniques.
>
>     - We would like to also highlight that our ATraDiff's interaction with the environment is deliberately constrained in our method. As detailed in Section 4.2, the environment access probability is set as $\frac{\rho}{(1-\rho)L}$. Here $L$ represents the expectation length of generation, $T$ represents the real sample time, and $\rho$ is the controlling variable. This ensures that environment interactions with ATraDiff occur only $\rho * T$ times, a constant and limited number, thereby controlling the potential increase in time costs.
>
>     - We appreciate the reviewer’s suggestion to train a reward prediction function from an offline dataset. To achieve this, we have followed the established methodology from prior work [Ref1], and implemented supervised training with the offline dataset. The effectiveness of this reward predictor has been demonstrated in our offline experiments, the results of which are included in Section C.1 of the updated appendix as well as shown in the table below. In our future work, we aim to adapt this reward predictor for online scenarios and further improve the performance.
>
>     |Task Name|TD3+BC|TD3+BC+ours_with_learned_reward|IQL|IQL+ours_with_learned_reward|
>     |---|:---:|:---:|:---:|:---:|
>     |halfcheetah-random|11.3 | 11.7 | 15.2 | 16.8|
>     |halfcheetah-medium|48.1 | 49.3 | 48.3 | 50.2|
>     |halfcheetah-medium-replay|44.8 | 45.3 | 43.5 | 45.6|
>     |halfcheetah-medium-expert|90.8 | 91.9 | 94.6 | 94.5|
>     |hopper-random|8.6 | 14.8 | 7.2 | 8.3 |
>     |hopper-medium|60.4 | 64.5 | 62.8 | 72.2|
>     |hopper-medium-replay| 64.4 | 65.2 | 84.6 | 97.0|
>     |hopper-medium-expert| 101.1 | 108.7 | 106.2 | 108.3 |
>     |walker-random| 0.6 | 3.4 | 4.1 | 4.0 |
>     |walker-medium|82.7 | 85.0 | 84.0 | 85.3 |
>     |walker-medium-replay|85.6 | 87.5 | 82.6 | 84.9|
>     |walker-medium-expert|110.0 | 111.1 | 111.7 | 111.6 |
>
>
> 2. Quality of offline datasets
>
>     - We acknowledge that the quality of the offline dataset impacts overall performance in reinforcement learning. However, we would like to emphasize that **our work introduces a versatile tool for online reinforcement learning that operates effectively regardless of data quality**.
>
>     - In our experiments, we observe that 1) high-quality data significantly increase the performance gap between our method and the baseline; 2) importantly, our method is designed to be robust and it does not require exceptionally high-quality data to function effectively, as shown in the experiment with medium-quality data in Section C.3; 3) our method is comparable to or better than the baseline with low-quality data; and 4) even datasets with non-optimal trajectories are sufficient for our method to achieve reliable performance, as shown in the multi-task experiment in Fig. 5.
>
>     - Regarding the experiments on D4RL, in the original manuscript, we followed prior studies [Ref2] and presented the average performance across different datasets from the D4RL benchmark. Following the reviewer’s suggestion and to provide a more comprehensive understanding, we have now included separate results for datasets of varying quality in Section C.3 of the updated appendix. The results in C.3 show how data quality impacts the effectiveness of our method and baselines across a spectrum of scenarios – while our method is comparable to or better than the baseline with low-quality data (like random data), the performance gap between our method and the baseline becomes much more pronounced with medium/high-quality data (like medium-expert data).

---

> ### Author Response · Authors · 2023-11-23
> **Response to Reviewer GZoQ (Part 2)**
>
> 3. Unfair comparison with pure online methods
>
>     - We would like to clarify that **our comparison is fair, because both our method and the baselines use the same offline datasets**. Specifically, as mentioned in Section 5.2 of the original manuscript, our comparison primarily focuses on **offline-to-online** methods like SPiRL, RLPD, and PARROT, which, like ours, leverage offline datasets in conjunction with online learning. The distinction between our method and these baselines lies in our unique utilization of offline data. Therefore, our improvement indeed comes from our methodology.
>
>     - In addition, the reviewer suggested using learning from demonstration methods as baselines. However, such methods usually require high-quality and task-specific trajectories as a training dataset, which may not achieve good performance with low-quality datasets and would be worse than the offline-to-online baselines we compared with.
>
>     - Regarding the specific ways of utilizing offline data, the Related Work section details how our method significantly differs from these existing offline-to-online methods.
>
> References:
>
> [Ref1] Ksenia Konyushkova, Konrad Zolna, Yusuf Aytar, Alexander Novikov, Scott Reed, Serkan Cabi, and Nando de Freitas. Semi-supervised reward learning for offline reinforcement learning. NeurIPS Workshop, 2020.
>
> [Ref2] Philip J. Ball, Laura M. Smith, Ilya Kostrikov, and Sergey Levine. Efficient online reinforcement learning with offline data. ICML 2023.

---

### Official Review · Reviewer_Kzne · 2023-11-10

**Soundness:** 4 excellent
**Presentation:** 3 good
**Contribution:** 4 excellent
**Rating:** 6
**Confidence:** 4

**Summary:**

"ATraDiff: Accelerating Online Reinforcement Learning with Imaginary Trajectories" addresses the challenge of training autonomous agents with sparse rewards in online reinforcement learning (RL). The authors propose a novel approach called Adaptive Trajectory Diffuser (ATraDiff), which uses offline data to learn a generative diffusion model. This model generates synthetic trajectories that serve as data augmentation, thereby enhancing the performance of online RL methods.

The key advantage of ATraDiff is its adaptability, which allows it to handle varying trajectory lengths and mitigate distribution shifts between online and offline data. The simplicity of ATraDiff enables it to integrate seamlessly with a wide range of RL methods. Empirical evaluations show that ATraDiff consistently achieves state-of-the-art performance across various environments, with significant improvements in complex settings.

The paper also discusses related work in the areas of offline pre-training for online RL, diffusion models in RL, and data augmentation in RL. It provides a background on Markov Decision Processes (MDPs), diffusion models, and replay buffers, which are essential components of the proposed method.

The methodology section of the paper details the design and training of ATraDiff, its deployment in online RL, and an online adaptation mechanism that addresses distribution shifts. Experiments demonstrate the effectiveness of ATraDiff in improving the performance of online RL methods and offline-to-online RL algorithms in complicated environments.

**Strengths:**

A solid and technically sound paper. Addresses an important problem in RL using generative (diffusion) models for generating imaginary trajectories. Generality is another strength of the method -  ATraDiff can be integrated with any online RL algorithm with a replay buffer. Authors address in a clever way varying trajectory lengths and potential distribution shifts. The efficiency of the method was confirmed by a diverse set of experiments and ablation studies.

**Weaknesses:**

There is no data about trade-offs associated with using ATraDiff. Diffusion model data generation can be pretty slow. How does it affects wall-clock training time?

**Questions:**

Diffusion model data generation can be pretty time-consuming. How ATraDiff affects wall-clock training time for SAC and REDQ? It would be good to see more discussions and data on this topic.

---

> ### Author Response · Authors · 2023-11-23
> **Response to Reviewer Kzne**
>
> Thanks a lot for your insightful and inspiring comments! We provide the following clarifications in response to your concerns:
>
> 1. Lack of wall-clock training time in experiments
>
>
>     - We acknowledge the concern regarding the increased wall-clock training time associated with the use of diffusion model data generation in ATraDiff. ATraDiff is designed with a trade-off between sample efficiency and computational resource demands. We would like to emphasize that **in the context of online reinforcement learning, the most significant cost often lies in interacting with the real environment and data collection**. Consequently, our primary focus with ATraDiff has been on optimizing sample efficiency to minimize the time required for environment interaction.
>
>     - Following the reviewer’s suggestion, we quantify the time impact of ATraDiff, by recording the total training time during experiments on the D4RL Locomotion dataset, running on a single NVIDIA RTX 4090TI GPU. The results are shown in the table below, which are also included in Table 4 of the updated Appendix. The comparison between SAC/REDQ and SAC/REDQ with ATraDiff shows that integrating ATraDiff with RL methods leads to an extension of the training duration. However, *this is counterbalanced by the improved efficiency in data utilization*.
>
>     ||walker2d|hopper|halfcheetah|
>     |---|---|---|---
>     |SAC|6.5h|7h|7h|
>     |SAC_with_ATraDiff|15h|17h|16.5h|
>     |REDQ|6h|6.5h|6.5h|
>     |REDQ_with_ATraDiff|14.5h|16.5h|16h|
>
>     - Moving forward, we are committed to enhancing the time efficiency of ATraDiff. Our goal is to reduce both training and inference times, while maintaining its current performance levels. We appreciate the reviewer’s valuable feedback and view it as a guiding direction for our future work.

---

### Author Response · Authors · 2023-11-23
**General Response**

We are deeply grateful for the time and effort all the reviewers have invested in reviewing our manuscript. The reviewers’ constructive suggestions have been invaluable in enhancing the quality of our work. We particularly appreciate reviewers’ remarks acknowledging the strengths of our work:

- (Kzne, GZoQ, 8Lyo, rtc7) Using diffusion model to generate trajectories as data augmentation approach is novel.
- (Kzne, GZoQ, cnq2, 8Lyo) ATraDiff can improve the data efficiency and performance of any online RL algorithm as a plug-in method.
- (cnq2, rtc7) The approach shows promising results across a wide variety of datasets and benchmarks.

In response to the insightful questions raised, we have provided detailed clarifications in the direct responses. These elaborations aim to address the reviewers’ concerns comprehensively. We hope that these clarifications and additional experimental results resonate with the reviewers’ expectations and adequately address the issues highlighted.

In addition, we have updated our manuscript including the appendix. In particular, we have provided additional explanations and experimental results on the following:
- Comparisons between our method and SynthER, validating our superiority over SynthER.
- Additional ablations, such as on the random dropping strategy and different prompts.
- Wall-clock running time of our method.
- Effectiveness of our method with learned reward prediction function.
- Visualization of the generated trajectories.
- A diagram figure (Fig. 2) that illustrates our framework, to facilitate the understanding of our algorithms and their underlying processes.
- Thoroughly updated appendix to supply more information, adding more algorithm and experiment setting details, and sections about the new experiments.

We remain open to further discussion and would appreciate any additional feedback or clarification requests the reviewers might have. Reviewers’ insights are instrumental in guiding the improvement of our manuscript.

Once again, thank you for your valuable comments and suggestions.

---

### Meta-Review · Area_Chair_oUo4 · 2023-12-05

**Metareview:**

This paper proposes ATraDiff, which upsamples an offline dataset to generate experience for an RL agent. Results show ATraDiff can boost online RL performance, however it has been noted by several reviewers that this work is remarkably similar to a recent paper SynthER. The authors claim the main difference here is the use of an offline dataset for online RL, but this seems to be a small change vs. the SynthER paper which decomposes these as two separate problems. In addition, the authors claim this NeurIPS paper was not published on time to be a baseline but having searched it appears this paper was on arxiv in March 2023, six months before the ICLR deadline. For that reason, the paper should be re-written to provide clear incremental progress and novelty over SynthER.

**Justification For Why Not Higher Score:**

The work seems incremental vs. SynthER and the authors justification that this was published at NeurIPS is not valid since the paper was on arxiv in March 2023. The work should be reframed in terms of the novelty wrt that method. In addition, the reward generation component could be limiting given rewards are not provided from large Internet scale datasets.

**Justification For Why Not Lower Score:**

N/A

---

### Decision · Program_Chairs · 2024-01-16

Reject